# Light-mediated communication in responsive materials ranging from individual self-oscillators to feedback-driven network

Hongshuang Guo[1], Kai Li [2] ✉, Jianfeng Yang [1], Dengfeng Li[1], Fan Liu[1] & Hao Zeng [1] ✉

The natural interactive materials under far-from-equilibrium conditions have significantly inspired advances in synthetic biomimetic materials. In artificial systems, there are means of interaction between individuals, for instance, mechanical contact, hydrodynamic coupling, thermal gradient, chemical diffusion and magnetic field. However, they generally lack high directionality or sufficient interaction ranges. Here, we present a method for constructing highly directed, interactive structures via optical feedback in light responsive materials far from their thermodynamic equilibria. We showcase a photomechanical operator system comprising a baffle and a soft actuator. Positive and negative operators are configured to induce light-triggered deformations, alternately interrupting the passage of two light beams in a closed feedback loop. The fundamental functionalities of this optically interconnected material loop include homeostasis-like self-oscillation and signal transmission from one material to another via light. Refinements in optical alignment allow remote sensing and feedback networks capable of adaptation in both materials' shape-morphing states and oscillating frequencies. The results show a versatile design method for light-mediated interaction among responsive materials, with applicability in everyday materials.

The exploration of bioinspired material systems has evolved from static functionalities[1–3], such as structural colors, roughness, and high strength, towards embracing heightened levels of stimuli-responsive dynamism[4–6]. This paradigm shift encompasses mechanisms such as self-regulation[7], feedback[8], entrainment[9], and synergy[10], reflecting the adaptive characteristics observed in living organisms. To fully leverage the intricate dynamism akin to life, a roadmap is necessary to delineate the internal feedback mechanisms between the stimulus source and material responsiveness[11–13], as well as their external interactions and communications across different entities[14].

In a biological context, communication denotes the interactive behavior wherein one organism influences the present or future actions of another. Beyond communication methods reliant on sophisticated biological components (*e.g.* vision-based)[15], there exist several basic forms of communication of particular interest to reductionists. The first involves mechanical contact, exemplified by species of army ants forming mechanical structures for group transport[16]. Second is hydrodynamically cooperative cilia generating, for instance, flagella synchronization and metachronal waves in liquids[17]. The third occurs through the exchange of biochemical substances, as beetles and flies utilize pheromones for mating[18]. The fourth form operates via contactless means, illustrated by phenomena like bird echoes and the synchronized flashing of fireflies[19]. In the realm of bioinspired material research, the utilization of out-of-equilibrium[20] soft matter has enabled versatile life-like functions[21,22] and novel opportunities for mutual interaction between entities[14,23–26]. These have spurred the emergence of new developments across broader disciplinary perspectives, including mechanical intelligence[27,28], microrobot swarm[29,30],

[1]Faculty of Engineering and Natural Sciences, Tampere University, P.O. Box 541 Tampere, Finland. [2]Department of Civil Engineering, Anhui Jianzhu University, Hefei, China. ✉e-mail: kli@ahjzu.edu.cn; hao.zeng@tuni.fi

collective matters[31–33], and systems chemistry[34–36]. Life-like matters operated far from equilibrium underscore the autonomy and adaptive nature of interactive constructs, bringing new trends in research grounded in dissipative mechanisms[37]. Current interactions in synthetic materials hinge on the localized interplay between physical and chemical variables, such as mechanical nonlinearity[38] and concentration diffusion in chemical feedback reactions[39,40]. Frequently, these interactions entail diminished spatial transfer distances or temporal delays. An approach facilitating low temporal delay, spatial coverage across long distances, and precise directional control offers promising potential for realizing life-like artificial systems with programmable interaction.

Here, we propose that photomechanical self-oscillators[41–43] can serve as elementary units for light-mediated communication between individuals, free of spatial restriction. A self-oscillator[44] is a structure capable of self-exciting and sustaining its mechanical motion by absorbing energy from a constant field, operating far from thermodynamic equilibrium. Typically, self-oscillators involve negative feedback[45], establishing an internal homeostasis-like steady state that exhibits resistance to external disturbances and is self-regulated by the materials' stimuli-responsiveness[13]. In this study, we demonstrate the coupling of self-oscillators through external light beams, enabling light-mediated interaction between two materials via a closed feedback loop involving negative and positive operators (see Fig. 1a). Additionally, we unveil the ubiquity of stimuli-responsiveness and extend the method to common materials encountered in everyday life.

## Results

### Optically coupled oscillators

The predominant method for creating a photomechanical self-oscillator involves utilizing a self-shadowing mechanism[46]. A light beam induces material deformation, which subsequently shields the incident light, resulting in the cessation of the deformation itself (negative feedback). By autonomously controlling the activation and deactivation of light on the responsive material, oscillation is achieved. In our approach, we employ a baffle-actuator system, as depicted in Fig. 1b, to separate the light-active and light-shielding components. A liquid crystalline elastomer (LCE) exhibiting reversible light-induced bending serves as the photomechanical element[47]. The sample was prepared with a splayed alignment, resulting in bending deformation upon thermal stimulation. When exposed to light, the dyes within the LCE convert photon energy into heat, causing the strip to bend toward the contracting surface (planar alignment), regardless of the direction of incidence. Detailed information regarding the chemical structures

and preparation process can be found in Supplementary Figs. 1, 2. For thermally and photothermally induced bending, temperature-induced strain, and mechanical properties, see Supplementary Fig. 3.

To shield the light, a lightweight aluminum foil is affixed to the end of the LCE actuator. Assembly instructions are provided in Supplementary Fig. 4. By adjusting the position of the light beam relative to the baffle, operators with contrasting functionalities can be programmed. As shown in Fig. 1c, when the initial beam is positioned below the baffle edge, the activated LCE moves towards the beam spot, tends to obstruct the light, thus acting as a negative operator. Conversely, when the baffle edge initially blocks the beam, the actuation moves the baffle away from the beam spot, allowing light to propagate, and the system functions as a positive operator (Fig. 1d).

To implement negative feedback in a single operator, a light beam is directed towards a mirror, reflecting onto the negative operator. The light induces deflection, prompting the baffle to interrupt the input beam. Upon cessation of light excitation, the material relaxes, permitting light beam propagation. Subsequently, the system reverts to its initial state, initiating a new cycle (Fig. 2a). A power threshold is required to deform the LCE and move the baffle closer to the laser spot, thereby initiating the negative feedback mechanism. Beyond such threshold, the operator undergoes self-oscillation fueled by a continuous light beam, with the amplitude increasing along the input power, as illustrated in Fig. 2b. The period ($T$) is determined by the resonance of the cantilever system[48]

$$T = \frac{2\pi}{\omega_0 \sqrt{1 - \frac{\bar{\beta}^2}{4}}} \tag{1}$$

where $\bar{\beta} = \beta\sqrt{l^3/3m\Pi}$, $\beta$ is the damping coefficient, $\Pi$ is the bending stiffness, $m$ is the mass of the baffle, $l$ is the length of the LCE cantilever, $\omega_0 = \sqrt{\frac{3\Pi}{ml^3}}$ is the natural angular frequency. Further details of the modeling are provided in Supplementary Note 2.1. In this basic self-oscillation mode, the periodicity remains insensitive to the fuel power, aligning with other self-oscillators based on the self-shadowing effect[46,49]. Conversely, positive feedback leads to an all-in or all-out bistable state. Detailed explanation of positive and negative feedback mechanisms in this model systems, see Supplementary Fig. 5.

To couple the operators, two laser beams are employed to establish a link between the negative and positive operators, as conceptually depicted in Fig. 1a. In the experimental setup, two operators

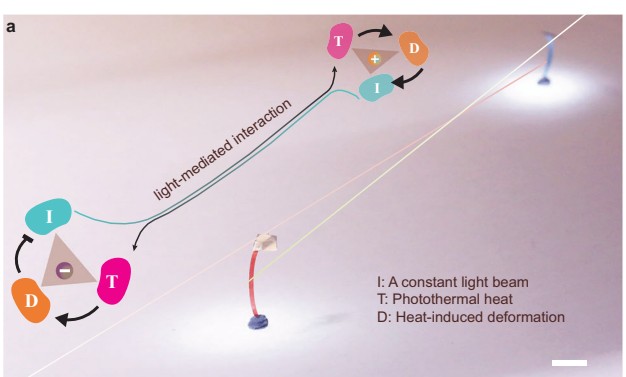

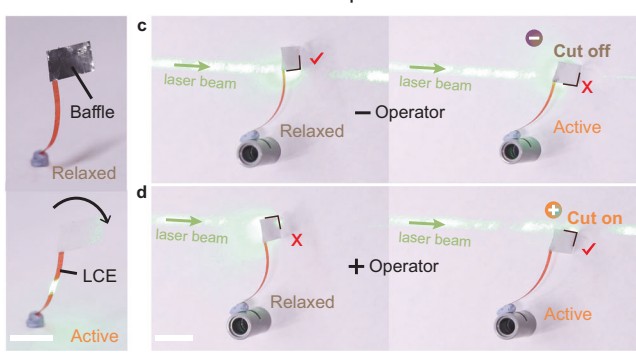

Photomechanical operators

**Fig. 1 | System concept. a** Schematics illustrating light-mediated interaction between two responsive polymers via a feedback loop. Inset is illustration of feedback, →, induce, –|, prohibit. The green line and red line indicate the propagation of two laser beams. **b** Photographs showing a baffle-actuator system in the relaxed state (top) and deformed state upon light illumination (bottom). The arrow

indicates the bending direction of the LCE strip upon actuation. Schematics of baffle-actuator systems depicting the (−)operator (**c**) and (+)operator (**d**) that cut off or cut on the light beam when actuated. Relaxed/active (−)operator passes/blocks the light; Relaxed/active (+)operator blocks/passes the light.

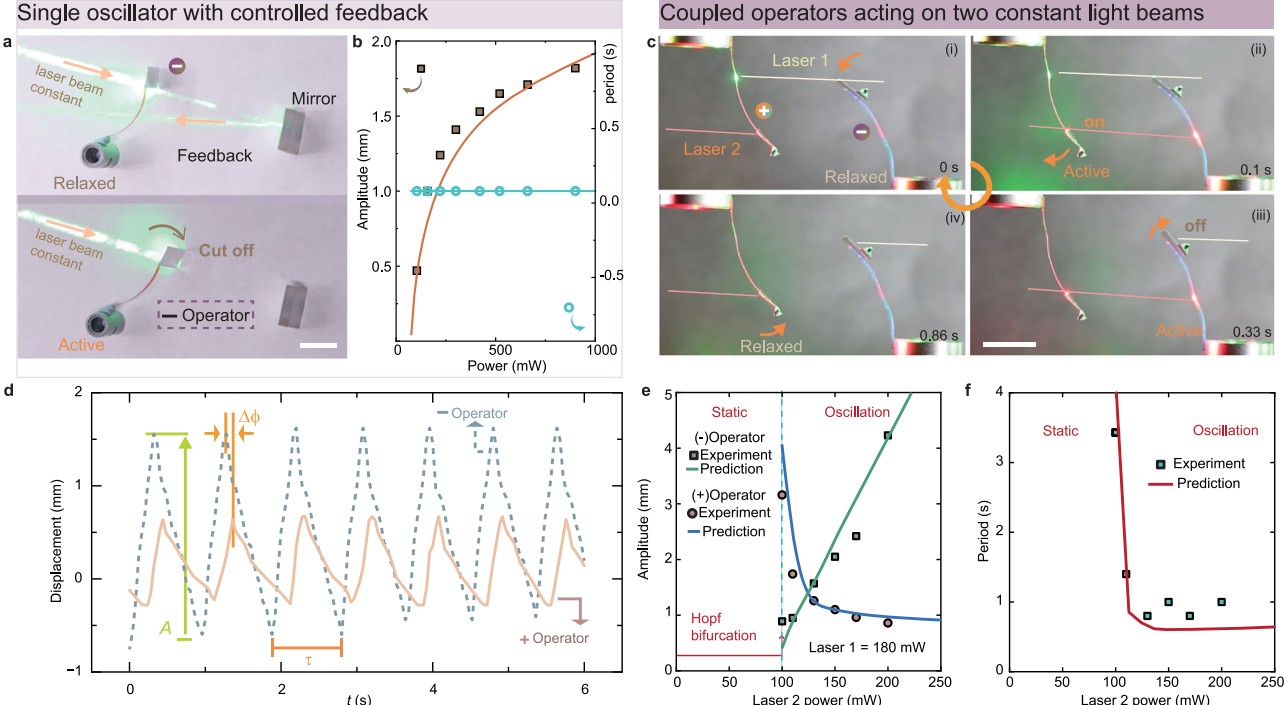

**Fig. 2 | From single self-oscillator to coupled oscillators. a** Schematics illustrating an optical feedback system through mirror reflection. The actuator is activated by the reflected laser beam that is controlled by the operator itself. Top, relaxed state (light on). Bottom, active state (light is cut off). **b** Variation of amplitude and period of single oscillator upon increase of excitation power. Light is 2 mm in diameter, 532 nm continuous laser. Solid lines represent the simulation results. **c** Photographs of coupling between a negative and a positive operator through two laser beams. The arrow indicates the instant deformation direction of the actuator. The red line indicates a red laser for excitation on blue-colored LCE and, a green light for a green laser on red-colored LCE. Red laser is 635 nm, 210 mW. Green laser is 532 nm, 138 mW. **d** Oscillation data of the coupled oscillators.

Displacement shows the change of tip positions of the two baffles. Insets show $\tau$, oscillation period, $A$, amplitude, $\Delta\phi$, phase difference between two operators. Laser 1 (532 nm) is 138 mW. Laser 2 (635 nm) is 210 mW. **e** Experimental and simulated results show the change of oscillation amplitudes of two operators upon varying the excitation power of the red laser beam. Laser 1 is 180 mW. **f** Experimental and simulated results showing the variation of the period with excitation power. Laser 1 is 180 mW. LCE sample dimensions are 24 × 2 × 0.1 mm³, baffle size is 5 × 20 × 0.01 mm³. All scale bars are 1 cm. Spot size is 2 mm (532 nm), 3 mm (635 nm). Details of the modeling and fitting parameters are provided in Supplementary Notes 2.1 and 2.2.

are positioned facing each other, as shown in Fig. 2c. Initially, beam 1 approaches the edge of the (−)operator, while beam 2 is obstructed by the (+)operator (step i). When beam 1 interacts with the (+)operator, it unblocks beam 2 (step ii). Subsequently, beam 2 activates the (−) operator, blocking beam 1 (step iii). Consequently, the (+)operator returns its position, blocking beam 2 (step iv), while the (−)operator returns to its original position, unblocking beam 1. The system reverts to state (i), initiating a new cycle. Additional information regarding the structural assembly is available in Supplementary Fig. 4c, while the oscillation kinetics are depicted in Supplementary Fig. 6 and Supplementary Movie 1. It is worth noting that two colors shown in Fig. 2c are used to distinguish between the laser beams and actuators—the red laser activates the blue LCE, while the green laser activates the red LCE, providing a clearer illustration of the working principle. The coupling mechanism does not depend on the excitation wavelength. In Figs. 3–5, only green laser beams (and red-colored LCEs) will be used for the experiments.

The coupling between two operators presents a distinctive case of self-oscillation. As illustrated in Fig. 2d, two operators become entangled, demonstrating identical periods and a phase shift. Interestingly, the coupled oscillator exhibits an extended period compared to individual ones. To elucidate this phenomenon, we developed a nonlinear dynamics model for two coupled oscillators and conducted numerical simulations (see Supplementary Method 2.2 for modeling details). Our modeling results reveal that the frequency of the coupled oscillator is no longer dictated by the resonance of the individual oscillator but is influenced by the delay in the material's light-response. The period of

the coupled system is derived and estimated to be

$$T \approx -4\tau_{\text{heat}} \ln\left(1 - \frac{\overline{w}_0}{\overline{P}}\right), \qquad (2)$$

where $\tau_{\text{heat}}$ is the characteristic time for heat exchange between photothermally-responsive LCE cantilever and environment, $\overline{w}_0 = \frac{w_0}{l}$ is the dimensionless on/off transition critical deflection of LCE cantilevers, and $\overline{P}$ is the dimensionless limit deflection of the LCE cantilevers under longtime illumination. The temporal profiles of two oscillators relative to the on/off states of the dual beams are illustrated in Supplementary Fig. 7. The delay, denoted as $t_d$ in Supplementary Fig. 7, represents the time required for the two baffles to transit alternately at their on and off positions. This delay is also reflected in the time gap between the state changes (on/off) of the two laser beams. The comparison between the single oscillator and coupled oscillator and the equivalence model for kinetics illustration, see in Supplementary Note 2.3.

By increasing the input power above a critical value, the system loses stability and transitions from the static state into self-oscillation, giving rise to Hopf Bifurcation as shown in Fig. 2e. Augmenting the power of one beam leads to an increase in the amplitude of the corresponding operator directly stimulated by that beam, while concomitantly diminishing the amplitude of the other operator responsible for modulating the ON/OFF state of the beam (Fig. 2e). It is notable that both laser beams induce deformations and are simultaneously regulated by their respective opposing operators. Alterations

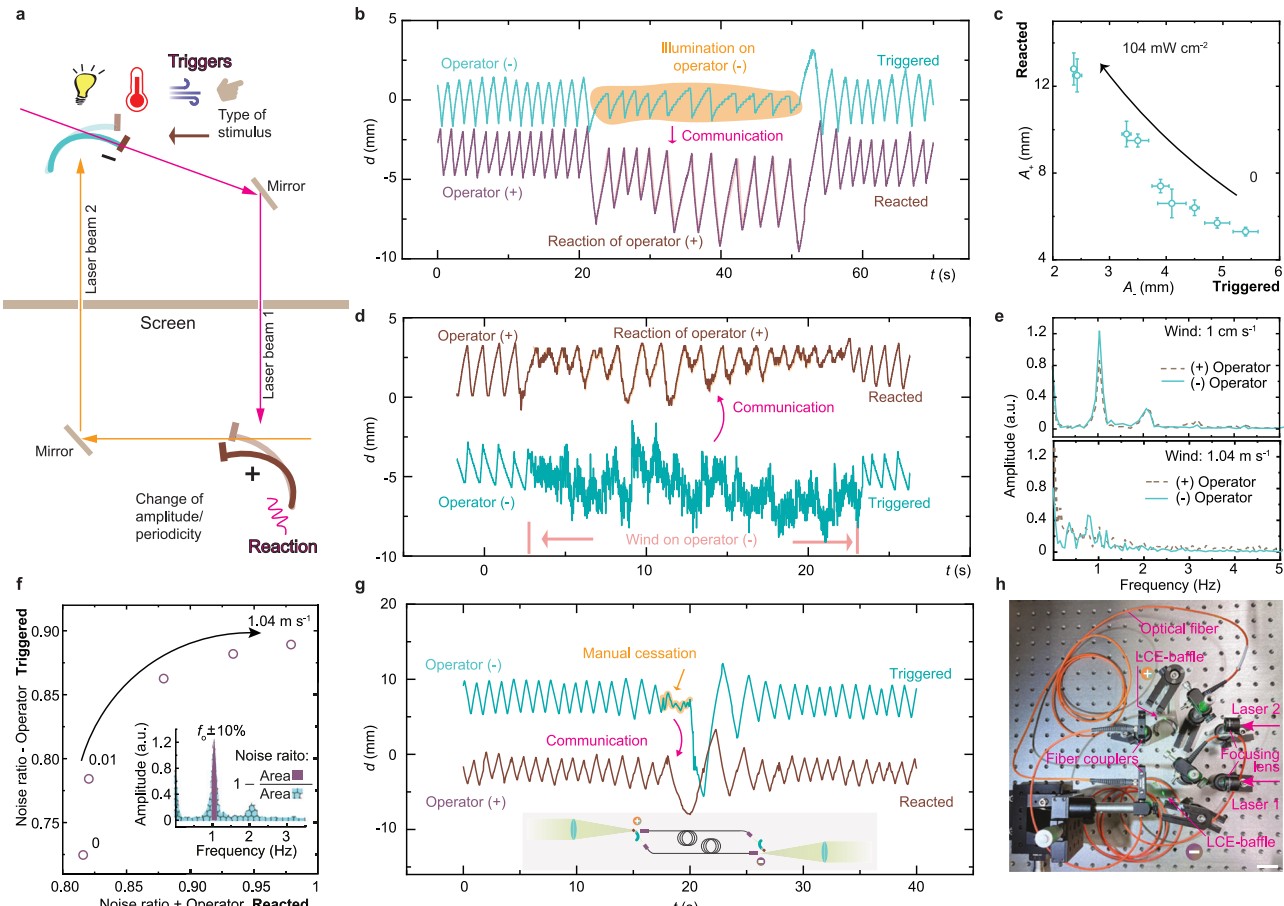

**Fig. 3 | Signal transmission between coupled oscillators. a** Schematic diagram of the signal exchange between two oscillations physically isolated by a screen barrier. Types of stimuli onto one of the operators are LED light, heat, wind, and manual stoppage. **b** Oscillation data of the coupled oscillators upon external light disturbance onto (−)operator. External LED light trigger is 635 nm, 69 mW cm⁻². **c** Amplitude coherence between two oscillators. Externally triggering light is 635 nm LED light source, from 0 to 104 mW cm⁻². **d** Oscillation data of the coupled oscillators upon wind disturbance onto (−) operator. Wind speed is 1.04 m s⁻¹. **e** Fourier transform of oscillation data upon different wind speed disturbances.

Light beams for coupled oscillators in (**b**−**e**) are 532 nm; laser 1, 540 mW, 3 mm spot size; laser 2, 70 mW, 1.2 mm spot size. **f** Noise ratio coherence between two coupled oscillators. Wind disturbance varies from 0 to 1.04 m s⁻¹. The inset depicts the definition of noise ratio. **g** Oscillation data of coupled oscillators based on optical waveguiding. The inset depicts the schematic diagram of the assembly of operators and optical fibers. Laser 1 power is 256 mW, Laser 2 power is 200 mW, laser beam are all 532 nm. The size of the light spot for laser 1 is 3 mm, and for laser 2 is 2 mm. **h** Photograph of the fiber-guided coupled oscillator system. LCE sample dimensions are 24 × 2 × 0.1 mm³, baffle size is 5 × 20 × 0.01 mm³. Scale bar is 2 cm.

in the intensity of either beam yield a predictable pattern of amplitude/period changes (Supplementary Fig. 8).

The reduction in oscillation period is attributed to the faster deflection of the actuator under higher laser power, resulting in expedited cutoff action and thus a shorter delay ($t_d$), thereby shortening the oscillation cycle. The experimental data qualitatively corroborates the theoretical predictions, as depicted in Fig. 2f and Supplementary Fig. 9. Two primary features of the system's dynamic behavior are observed. Firstly, period and phase synchronization between the two oscillators, indicating interdependence where cessation of one oscillator halts the motion of the other (Supplementary Fig. 10). Secondly, the system's overall behavior becomes responsive to local environmental changes—variations in beam power or added mass to either operator influence amplitude and frequency (Supplementary Figs. 11, 12). These characteristics offer a unique opportunity for signal transmission between materials via light beams. The distinguishing characteristics between single self-oscillators, interactive oscillators documented in the literature[23,24], and self-oscillators with light-mediated interaction presented in this study are summarized in Supplementary Fig. 13.

## Signal transmission

We established an optical feedback loop by physically isolating the material components using a screen board, as depicted schematically in Fig. 3a and Supplementary Movie 2. To facilitate signal transmission, we applied various triggers to one of the operators and monitored the resulting behavioral changes in the other. In the experiment, we applied four types of external triggers – temperature variations and LED light irradiation to induce additional bending of the soft actuator, mechanical contact to cease oscillations, and wind disturbance to enhance system randomness. Figure 3b illustrates the oscillatory behavior of the system in response to an external disturbance caused by LED light irradiation. Upon application of additional irradiation, the interrupted operator exhibits a decrease in oscillation amplitude, whereas the isolated one displays an increase in amplitude (compared with Fig. 2e). Upon cessation of LED irradiation, both operators return to their respective oscillation patterns. Hence, these two materials exhibit interaction with a negative correlation (Fig. 3c). Further details regarding oscillation under varying intensity triggers are provided in Supplementary Fig. 14. Interaction induced by heat disturbance is depicted in Supplementary Fig. 15.

In Fig. 3d, airflow induces stochastic oscillations in the directly affected operator, while the feedback loop transmits this randomness as a noise signal to the other. The degree of oscillatory irregularity of the system is quantitatively assessed through Fourier transform spectra, as illustrated in Fig. 3e. Under calm or mildly breezy conditions, both operators exhibit spectra with distinct peaks at their fundamental frequencies. However, as wind velocity increases, the amplitude peaks around the fundamental frequency diminish. For detailed oscillation data under varying wind speeds and associated Fourier transform spectra, please refer to Supplementary Fig. 16. In addition, we introduce a noise ratio metric, calculated as 1 minus the ratio between the spectral area at the fundamental frequency within a ± 10% bandwidth and the total spectral area encompassing all frequencies, as depicted in the inset of Fig. 3f. The noise ratio of the interconnected operators exhibits a positive correlation in their interaction (Fig. 3f), suggesting that heightened fluctuations in one operator, caused by wind perturbations, result in increased noise in the other.

Laser beam coupling enables long-distance interaction. Supplementary Fig. 17 demonstrates light interaction between two materials positioned on separate tables, with a six-meter gap between them. Fiber optics offers a compact alternative for transmitting light energy or signals, enhancing durability and facilitating systematization. Figure 3h presents a proof of concept of material interaction using fiber optics, where light beams traverse two fibers. These beams are directed through the operators, entering the fibers via focusing lenses. The outputs of the fibers then illuminate the corresponding operators, as illustrated in the schematic inset of Fig. 3g and Supplementary Movie 3. Manual interruption or release of one operator leads to cessation or resumption of oscillation in the other (Fig. 3g).

## Adaptation

Responsive materials can self-adapt to external stimuli, and the concept of adaptation has been demonstrated in various forms. Here, we echo the expression by Fratzl et al. [50], who described adaptive materials as representing a higher level of responsiveness, "the stimulus-induced change in the material encounters a competing reaction and the output results from balancing the two via mutual feedback". In the following, we explore the potential of using optical feedback loops in interactive materials to program three distinct types of adaptivity—bistability, cascading transitions, and dual rhythms.

Conventionally, stimuli-responsive materials can change their shapes, as illustrated in Fig. 4a. For example, a thermally sensitive polymer strip bends when heated and returns to its original, unbent state after removing the heat. Here, we refer to adaptation as the ability of a material to change its state in response to external stimuli and maintain the altered state even after the stimulus is removed. To achieve bi-stable states, we design the system with a positive feedback loop, as positive feedback enables an "all-in" or "all-out" bistable mechanism[51]. Supplementary Fig. 18 illustrates this operator and the surrounding optical excitation pathways, where the LCE strip serves as a proof-of-concept gripping device. The key principle is such that the material's deformation is not directly powered by the stimulus but rather operates as a self-powered, bi-stable system that can switch between two stable states upon external disturbances. In other words, it is the bifurcation of the dissipative system that enables dynamical change of condition by self-organizing into different shapes. Figure 4b demonstrates an example of thermo-adaptation, which differs from the conventional principle of stimuli-responsiveness by emphasizing the material's ability to maintain its state after the stimulus is turned off. In this example, a gripper closes when the LCE strip bends and reopens when the strip unbends. Figure 4c depicts a scenario where the gripper, initially opens (with a baffle blocking the light beam so the LCE is inactive), closes when exposed to a heated object (80 °C). The

bending of the structure unblocks the light beam, delivering energy to the actuator within the positive feedback loop. As a result, the gripper remains closed even after the object's temperature cools to room temperature. Note that there are two stable states of a positive feedback-driven system, as they can be mutually switched by mechanical triggers. Figure 4d presents the opposite case, where the gripper starts in a closed state (with the baffle unblocking the light beam, bending the LCE strip). When exposed to a cold object (0 °C), the gripper opens, and the baffle blocks the light beam. After the object warms to room temperature, the light beam remains blocked, and the gripper maintains its open state.

A series of positive operators connected in a closed feedback loop can result in a cascading transition of material states. This means that switching the state of one operator can trigger a sequential shape-change in others until all operators complete their transition. Figure 4e illustrates an example of five positive operators coupled by five laser beams in a closed loop. The system exhibits two stable states—(1) all operators are open, allowing all the light beams to pass through, and (2) all operators are closed, blocking all beams. Figure 4f captures snapshots of the transition, starting with all operators in the open state. A mechanical trigger is applied to the first operator to switch it OFF, which sequentially drives the 2nd to 5th operators into the closed state (Supplementary Movie 4). Displacement data shown in the lower right corner of Fig. 4f highlights the kinetics of the shape-morphing process, revealing a time delay between the transitions of sequential operators. For more detailed displacement data during the cascading transition, see Supplementary Fig. 19. State (shape) transition can be seen as bifurcation of the system, which depends on the strength of the external stimulus, particularly the triggering period. Figure 4g shows that a triggering period of less than 13 seconds cannot induce a complete transition. In such cases, the recently switched operator reverts to its previous state, reversing the sequence. However, for a triggering period of 13 seconds or more, all operators switch to the closed state, and the system stabilizes in its new configuration. In this case, bifurcations signify a change in the network's morphology, characterized by the shape-switching along operators. The bifurcation diagram of the all-closed to the all-open state transition is also shown in Supplementary Fig. 20.

The examples above illustrate the concept of adaptation in bistable states, demonstrated through shape changes in a single operator and sequential shape morphing in spatially separated operators. Oscillation frequency serves as another physical metric for assessing the state's condition. In the following, we demonstrate that bifurcations signify a change in the system's behavior, i.e., transitions between oscillatory rhythms.

Figure 4h presents the optical design of a material system incorporating two negative feedback loops. A negative feedback loop induces homeostasis-like self-oscillation with a well-defined frequency (see Fig. 2b). The baffle on operator 1 contains two holes that act as distinct negative operators, controlling the passage of beam 1 & 2. Beam 1 is directly reflected back to operator 1, inducing a self-oscillating system in accordance with the principles illustrated in Fig. 2a. Beam 2 interacts with another operator, which modulates beam 3. Beam 3, in turn, excites operator 1, following the coupled oscillator mechanism outlined in Fig. 2c. Each feedback loop includes a single negative operator and produces stable self-oscillation frequency. A mechanical trigger allows switching between two oscillation rhythms, as schematically depicted in Fig. 4i and Supplementary Movie 5. Governed by the first feedback loop, the system self-oscillates at a high frequency (5.5 Hz), as shown by the photograph and displacement data in Fig. 4j, k. After switching to the second feedback loop, which couples two operators, the system exhibits a lower frequency due to the increased delay within the loop (see Eq. 2 and Fig. 2f). The displacement data showing different

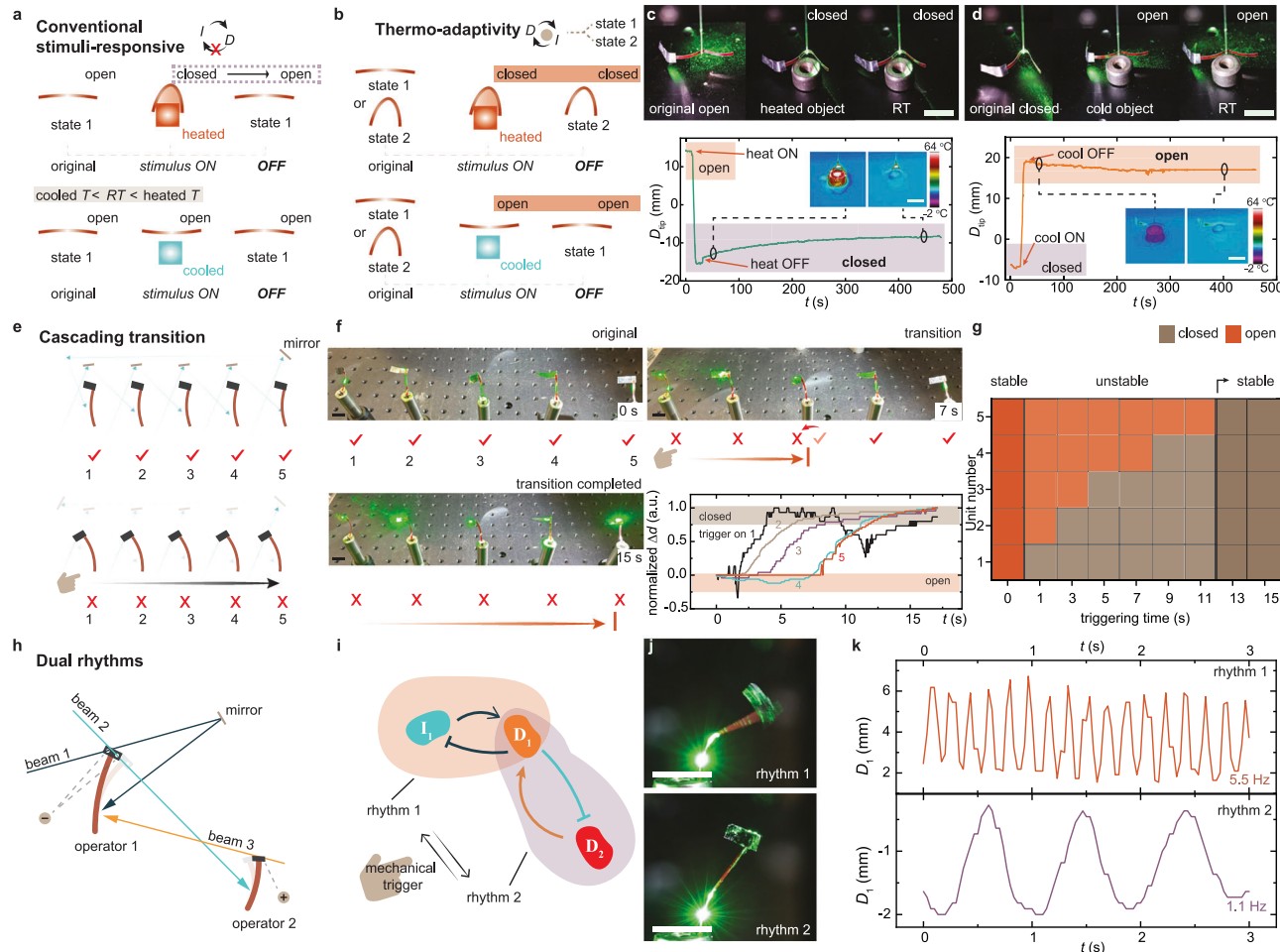

**Fig. 4 | Adaptation enabled by light-interactive network. a** Working principle of conventional stimuli-responsive materials. **b** Concept of material state is induced by a positive feedback loop, where adaptation refers to the change in state in response to external stimuli. **c** Top shows the photographs of an operator-based gripper initially in an open state, adapting to a hot object and changing to a closed state. Bottom shows displacement data of the actuator tip, $D_{tip}$, in response to heat from an 80 °C object. Insets are thermal camera images of the system before and after cooling to room temperature. **d** Top are photographs of an operator-based gripper initially in a closed state, adapting to a cold object and opening. Bottom is displacement data of $D_{tip}$ in response to cold stimulus from a 0 °C object. Insets are thermal camera images of the system before and after warming to room temperature. **e** Schematic diagram of the optical pathway in a network of 5 positive operators, showing two stable states (all-open and all-

closed). **f** Snapshots of five operators responding to a mechanical trigger on the first operator. Lower right is the change in baffle position (Δd) for each operator. **g** System bifurcation, with the state change of each operator recorded as different trigger intervals are applied to the first operator. **h** Design of a dual rhythmic self-oscillator based on two feedback loops. **i** Schematic of the feedback mechanism in the dual rhythmic system. $I_1$, intensity of beam 1. $D_1$, displacement of operator 1. $D_2$, displacement of operator 2. **j** Snapshots of the operator 1 oscillating at different frequencies. **k** Displacement data of the oscillator. Laser used in (**c**, **d**) is 532 nm, 220 mW, 2 mm. Laser used in (**e**–**g**) is 532 nm, 100 mW, 4 mm. **h**–**k** Beam 1 excited on operator 1 is 532 nm, 180 mW, 2 mm; beam 2 spot on operator 2 is 532 nm, 180 mW, 2 mm; beam 3 spot on operator 1 is 532 nm, 160 mW, 2 mm. All scale bars are 1 cm.

frequencies and transitions between rhythms are provided in Fig. 4k and Supplementary Fig. 21. In summary, the examples above illustrate the adaptation concept enabled by optical feedback loops in responsive materials. We demonstrated self-adaptation through shape changes and rhythm switching in response to external stimuli, as well as the system's capacity to maintain these altered states (shape or rhythm) after the stimulus is removed.

## Conceptual generalization

A single (−)operator forms the basis of the individual self-oscillator (Fig. 2a), while the coupling between (−) and (+) operators establishes a two-element network, facilitating materials interaction (Fig. 2c). Figure 5a illustrates the expansion of this system concept into a light-based interactive network with an increased number of units. This networking strategy follows a succinct design principle – each communicator constitutes one operator (+ or −), with light beams

facilitating connectivity between operators to expand the network. In Fig. 5a, a series connection of four units is depicted, accompanied by oscillation data recorded as shown in Fig. 5b. Similar to the behavior observed in the previous coupled network, manual cessation or activation at one unit triggers the cessation or reactivation of the remaining operators sequentially. For comprehensive oscillation data in the coupled network with varying unit numbers, please refer to Supplementary Fig. 22 and Supplementary Movies 6 and 7. Additionally, Supplementary Fig. 23 presents other regulatory network designs inspired by biochemical oscillators[52].

The light-induced deformation observed in LCEs serves as a specific example of stimuli-response, known as photomechanics, in synthetic materials. This phenomenon of stimuli-response is prevalent in both natural and human-made systems, many of which can be exploited to develop feedback mechanisms, establish networks, and facilitate interaction using light beams. Figure 5c illustrates a collection of everyday materials endowed with light responsiveness, categorized

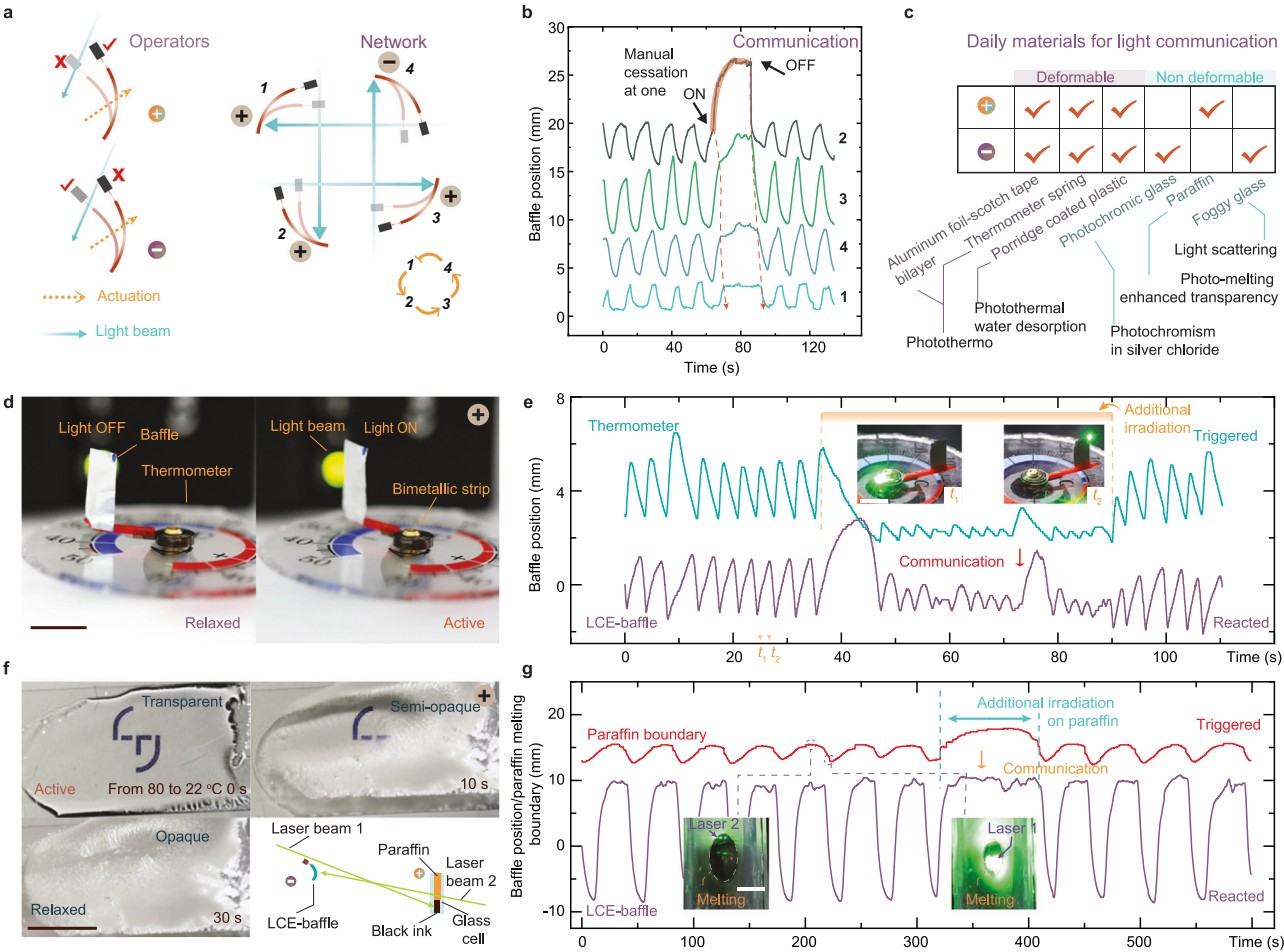

**Fig. 5 | Conceptual generalization. a** Left is the schematic diagrams illustrating the work-principle of individual operators. Dashed arrows indicate the actuation direction of the operator, while solid arrows indicate the propagation of the laser beam. Right is the design of a network composed of four coupled operators. **b** Oscillation data of a four-unit-coupling network. Dashed lines indicate the sequence of reactions transferred to the neighboring operators upon manual cessation. Four laser beams are 532 nm, 320 mW, 2 mm spot size. **c** List of daily materials built-in with stimuli-responsiveness. **d** Photographs of a (+)operator made of thermometer spring and aluminum baffle. **e** Oscillation data of the coupling between LCE-baffle and a thermometer spring. Insets are the snapshots of the thermometer deformations at different oscillation states. Laser excited on LCE is 532 nm, 43.6 mW, 2 mm spot size. Laser excited on thermometer is 532 nm, 930 mW, 3 mm. **f** Photographs of paraffin on glass substrate upon cooling from 80 °C. Insert is schematic diagram of the coupling between paraffin and LCE-baffle. **g** Oscillation data of the coupling between a paraffin plate and LCE-baffle. Insets show photographs of paraffin during oscillating. Laser excitation on LCE is 532 nm, 80 mW, 1.2 mm spot size. Laser excitation on paraffin is 532 nm, 1440 mW, 2 mm spot size. LCE sample dimensions are 24 × 2 × 0.1 mm³, baffle size is 5 × 20 × 0.01 mm³. All scale bars are 1 cm.

into deformable and non-deformable materials. Deformable materials include aluminum foil (commonly used for baking in ovens) and Scotch tape, capable of forming a bilayer structure that undergoes deformation upon temperature increase. This deformation is attributed to the differential (photo-)thermal coefficients between the two layers[53], akin to the operating principle of a thermometer spring. A porridge thin film demonstrates expansion upon absorbing water in humid environments. Upon photothermal heating, this film releases its water content, resulting in surface shrinkage toward the light source and inducing film bending. These deformable materials offer opportunities for constructing operators based on the principles depicted in Fig. 1. Another category involves materials capable of effectively controlling light propagation without the need for mechanical structures. For example, photochromic glass containing particles of silver chloride darkens upon exposure to UV light[54], serving as a negative operator. Paraffin, initially opaque at room temperature, turns transparent when heated, acting as a positive operator. Furthermore, when a glass piece is positioned over a container of water and the water is photo-heated, fog formation on the glass scatters light, rendering the entire system a negative operator. Supplementary Figs. 24–26 elaborate on

various stimuli-responsive concepts in everyday materials and the design principles underlying operator functionality. Subsequently, we briefly highlight an example from each category.

In Fig. 5d, the responsiveness of a thermometer spring is depicted, showcasing its function as a positive operator. The metallic spring rotates at a rate of 3° per Kelvin, and upon application of black ink, it responds to light through a photothermal effect. This action adjusts the position of a baffle atop its pointer, thereby modulating the light beam. Consequently, negative and positive feedback can be optically looped between an LCE-baffle (−operator) and a thermometer (+operator), as demonstrated by the previously illustrated principle. The coupled system demonstrates self-sustained oscillation and facilitates interaction, as depicted in Fig. 5e and Supplementary Movie 8. Additional data on disturbed oscillation resulting from external mechanical cessation can be found in Supplementary Fig. 27. For further details on optical alignment, sample preparation, and structural orientation, refer to Supplementary Fig. 28. Figure 5f showcases a positive operator based on the transparency modulation of paraffin. Above its melting temperature (approximately 80 °C), the paraffin becomes transparent to light, while upon cooling, it turns opaque,

hindering light propagation. To induce photothermal heating, black ink was incorporated into the paraffin, and the two laser beams were aligned to couple between an LCE-baffle and the paraffin. The optical setup is depicted in the inset of Fig. 5f. Figure 5g presents data on the coupled oscillations of the system, with the boundary position indicating the melting region of the paraffin – a higher position value corresponds to a larger melting area allowing light transmission, while a lower value indicates a smaller melting region resulting in light blockage. Insets in Fig. 5f display photographs depicting the corresponding states of the paraffin. The oscillation and interaction between the LCE and paraffin are facilitated by the self-regulation of light propagation between the baffle and paraffin. Further details on optical alignment, paraffin preparation, and additional on oscillation can be found in Supplementary Figs. 29, 30 and Supplementary Movie 9.

## System limitations

Unlike thermo-responsive cantilevers made of metallic bilayers, soft actuators composed of LCEs exhibit non-uniform deformation along the strip during dynamic oscillation. Supplementary Fig. 31 presents a series of photographs capturing the shape-morphing behavior of the soft LCE in the experiments, revealing a wide variety of configurations. These variations arise from differences in sample orientation and the position of the laser excitation spot relative to the strip in different experimental setups. Additionally, the light-heat-induced deformability varies among samples due to the self-assembled nature of liquid crystals—even strips cut from the same film prepared in the same cell can exhibit subtle differences in actuation behavior (Supplementary Fig. 3b). The viscoelastic properties of the soft material further contribute to a gradual drift in oscillation behavior over extended periods, as observed in the Supplementary Figs. 4, 10, 14, 16, 17, 29. Moreover, this light-mediated interaction mechanism depends critically on the precise alignment of the laser. Two focused laser spots must be positioned near the operators' baffles to enable the cut-on/off action. Details are seen in boundary conditions in the model, Eq. 11, 12 and Eq. 19, 20 in Supplementary Note. However, the baffle's critical distance $w_0$ between the cut-on/cut-off position and the system's equilibrium point, cannot be predetermined prior to the onset of dynamic self-oscillation. These factors collectively result in a variation of the delay time, $t_d$, in the coupled oscillator. Supplementary Note 2.3 shows theoretical prediction data, demonstrating that changes in $w_0$ influence the oscillation frequency, amplitude, and waveform.

It is noticed that light-mediated interactions in soft matter systems should not aim for precise control or fine-tuning of oscillatory properties across different samples. Instead, this study highlights the significance of achieving non-equilibrium states, enabling signal transmission and feedback-loop-driven adaptation, which are the key features inspired by biological systems.

The gripping and cascading devices illustrated in Fig. 4 serve to demonstrate adaptation at the proof-of-concept level. Their operation requires precise optical beam alignment and accurate positioning of the heat stimulus source within a specific range to overcome the energy barrier governed by the positive feedback loop. For broader applicability, system-level integration between the actuator and the optics will be necessary.

## Discussion

The self-shadowing effect has been widely employed in the development of LCE self-oscillators[11,46]. However, most implementations are limited to individual oscillators. Few studies have demonstrated interactions between self-oscillating strips via mechanical vibrations[23] or hydrodynamic forces[24], leaving significant potential for expanding interactive units[55,56]. Depending on the coupling strength, a disturbance to one oscillator can have varying effects on the other[57]. In this study, we present a method to achieve strong coupling between

two oscillators using light beams—such that if one oscillator ceases, the other also stops. This behavior is distinct from that observed in existing self-oscillating systems (see Supplementary Fig. 13). This method also offers designs of positive and negative operators that can be programmed to establish diverse feedback loops among multiple oscillators, offering alternatives for biomimetic research by emphasizing the central role of feedback in the realization of bioinspired functions.

Research on bioinspired materials has evolved from focusing on static functionalities to embracing increasing levels of dynamic responsiveness and interactive behaviors[4–6,31]. These advancements highlight the autonomy and adaptability of biological systems while inspiring the introduction of new interaction behaviors and functions in synthetic responsive materials. In this study, we report a method that couples two or more self-oscillators through an optical feedback loop, enabling highly directional network connections and facilitating material interactions over long distances with minimal delay. The feedback loop can selectively engage different oscillating elements, thereby allowing adaptation at the system level. It is noted that the feedback principle is built externally through the optical pathway, not relying on the specific design of internal material properties. The demonstrated principle holds promise for application to other responsive materials[20,58]. Hydrogels, for instance, typically exhibit a lower critical solution temperature (LCST), enabling light-heat-triggered opaqueness above the LCST. Zhang et al. employed this light-induced opaqueness as a negative operator to achieve self-oscillation in a laser-excited gel tube[13]. Similarly, photothermal bilayers, and humidity-sensitive actuators that can mechanically respond due to light-heat-induced desorption[59], undergo reversible deformation and thereby function as positive and negative operators, akin to the LCEs utilized in this study. Additionally, joule-heat triggered LCEs[14] and shape-memory alloys[60] have been used as positive and negative operators within feedback networks built upon electrical circuits, whose underlying mechanisms are comparable to the light-mediated approach demonstrated here. In previous literature, the emphasis has often been placed heavily on the material properties necessary to reach non-equilibrium states[13,61]. In contrast, we believe that stimuli-responsiveness is a widespread characteristic, as illustrated by everyday materials shown in Fig. 5. We advocate for greater interdisciplinary dialogue to establish a general design framework for communication in synthetic materials.

In previous literature, adaptation has often been described as the responsive shape change of a material in reaction to environmental stimuli. In this study, we propose a broader perspective – adaptation can be viewed as the capacity of a material to change its state in response to external disturbances, coupled with the ability to retain this altered state even after the disturbance is removed. Typically, maintaining an altered state requires mechanisms, e.g., shape memory effects[62], plastic deformation (as in deformable clays), or bi-stable mechanical architectures[63]. Here, in Fig. 4b–g, we demonstrate that a positive feedback loop can induce bi-stability without relying on shape memory, microstructural changes, or specific mechanical designs. Moreover, a material state can be characterized not only by its spatial configuration (the shape) but also by its oscillation frequency in the time domain[37]. Thus, adaptation may also refer to the ability to shift and sustain different oscillation frequencies following external disturbances, as illustrated in Fig. 4h–k.

For the conventional robotic control perspective, the light-mediated interaction concept provides a possibility for remote sensing and electrical control of soft actuators over long distances. Details see in Supplementary Note 2.4 Sensing and control.

A miniaturized cantilever inherently exhibits higher oscillation frequencies due to fundamental scaling laws. Downscaling the self-oscillators enables both higher device density and increased sampling rates. Two-photon polymerization is a well-established fabrication

technique for realizing microscopic LCEs and optical waveguides for efficient coupling. We envision scalability arising from the ever-increasing unit density and complexity of the feedback network. Further details are provided in Supplementary Note 2.5 Scalability—a future plan.

In this paper, we present a method for achieving light-mediated communication among non-equilibrium materials, free of any electronics. We introduced an experimental model consisting of a soft actuator and a baffle, where a photomechanical liquid crystalline elastomer actuator and aluminum foil determine the operator's function. Commencing with a single negative operator, we demonstrate the mechanism of optical feedback in self-oscillation. Subsequently, we couple a positive and a negative operator to establish a closed loop of optical feedback. We observe that the frequency of the coupled system depends on the delay of the material's light responsiveness, rather than the resonance of the single oscillator. This coupled feedback loop renders two materials interdependent in oscillation, facilitating signal transmission between them. We utilized manual triggers—mechanical contact, heat, light, and wind—to perturb the oscillation of one operator, which then autonomously conveyed the disturbance signal to the coupled operator through the feedback network. While our focus did not center on achieving quantitative control over material deformation via optical excitations, these experiments served as a conceptual illustration of primitive interaction between non-equilibrium materials using light. Such light-mediated interaction is untethered, with high directionality, and free of spatial restriction. We provided evidence supporting the systematic development of this concept, demonstrating examples of interaction over meter-scale distances, signal transmission via optical fibers, series connections with an increased unit number, and the design of complex networks inspired by biological oscillators. We emphasized the widespread occurrence of stimuli-responses in both natural and synthetic materials, exemplified by light-mediated interaction utilizing commonplace materials such as a thermometer spring and paraffin. We highlighted three adaptive activities that are enabled by using optical feedback loops, including bi-stable states, cascading transition and dual rhythm. This work presents an approach for enabling interaction between soft materials using light.

## Methods

### LCE Material preparation in brief

1,4-Bis-[4-(6-acryloyloxyhexyloxy) benzoyloxy]-2-methylbenzene (99%, RM82) was obtained from SYNTHON Chemicals. 6-Amino-1-hexanol and dodecylamine from TCI, 2,2-Dimethoxy-2-phenylaceto-phenone were obtained from Sigma-Aldrich, Disperse Red 1, and Disperse Blue 14 were obtained from Merck. ST06512 [4-(6-(Acryloyloxy) hexyloxy)phenyl 4-(6-(acryloyloxy)hexyloxy)benzoate] was obtained from SYNTHON Chemicals. GDA (tetraallyloxyethane) and DPA (dipropylamine) were obtained from TCI. EDDT [2,2′-(ethylenedioxy) diethanethiol], PETMP [pentaerythritol tetrakis(3-mercaptopropio-nate)], BHT [2,6-di-tert-butyl-4-methylphenol], Irgacure 651 [2,2-dimethoxy-2-phenylacetophenone], were obtained from Sigma-Aldrich. All chemicals were used as received.

### LCE strip actuator

Liquid crystal cells were made by adhering two glass substrates, one coated with a homeotropic alignment layer (JSR OPTMER, spun at 4000 RPM for 1 min, followed by baking at 100 °C for 10 min and then at 180 °C for 30 min), and the other with poly-vinyl alcohol (PVA) rubbed on (5% water solution, spun at 4000 RPM for 1 min, and baked at 100 °C for 10 min) for uniaxial alignment. Microspheres with a diameter of 100 μm (Thermo Scientific) were employed as spacers to determine the film

thickness. The liquid crystal mixture consisted of 0.3 mol RM82, 0.05 mol 6-Amino-1-hexanol, 0.05 mol dodecylamine, and 2.5 wt% of 2,2-Dimethoxy-2-phenylacetophenone, mixed at 95 °C. This mixture was introduced into the cell by capillary action at 95 °C and then gradually cooled down to 63 °C (at a rate of 1 °C per minute). The cell was then kept in an oven at 63 °C for 24 h to allow for an aza-Michael addition reaction for oligomerization. Subsequently, the sample underwent UV irradiation (360 nm wavelength, 180 mW cm$^{-2}$ intensity) for 20 min to initiate poly-merization. Finally, the cell was opened using a blade, and strips were cut from the film.

### LCE fiber actuator

The precursor mixture consisting of 0.36 mmol ST06512, 0.32 mmol EDDT, 0.04 mmol PETMP, 0.04 mmol GDA, 1.5 wt% Irgacure 651, 1.0 wt% BHT, and 0.5 wt% DPA was prepared and thoroughly homogenized at 80 °C. The mixture was then injected into silicone tubes and main-tained at 80 °C in an oven for 24 h to allow the thiol-Michael reaction to form oligomers. The resulting oligomers were mechanically stretched and subsequently polymerized under UV light (365 nm, 180 mW cm$^{-2}$, 20 min) to obtain fibers. Finally, the fibers were dyed in a Disperse Red 1-isopropanol solution.

### Daily materials sources

The materials used in constructing the bilayer experimental setup, namely aluminum foil and Scotch tape, were readily available from local stores. Oats and coffee, selected for their moisture-responsive and temperature-sensitive characteristics, were procured from the same outlet. Additionally, paraffin and a thermometer, for temperature-responsive behavior and thermal actuation, were also sourced locally.

### Sample assembly

To obtain LCE films of different colors and thus different light sensi-tivities, the prepared LCE samples were diffused with different dyes. This process entailed placing the samples on a hotplate set to 100 °C and uniformly dispersing powders of Disperse Red 1 and Disperse Blue 14 dyes onto the surface of the LCE, following a 5-minute thermal diffusion. The absorbance saturates around 2 after 4-minute diffusion, as seen in the spectra in Supplementary Fig. 32. After the diffusion, the residual powder on the surface was wiped away. The LCE films were then cut to dimensions of $2 \times 24 \times 0.1$ mm$^3$, and aluminum foil was also cut to dimensions of $5 \times 20$ mm$^2$. The cut aluminum foil which func-tioned as a baffle was bonded to one end of the LCE material using glue.

### Stimuli sources

Two continuous-wave lasers (at 635 nm and 532 nm wavelengths) served as excitation sources, while a CoolLED pE-4000 unit was employed for photopolymerization and served as an external light source for perturbing oscillations. Airflow was generated by a small fan, with wind speed regulated by adjusting the applied voltage and quantified using a Digital Hand-held Wind Speed Gauge Meter (GM816) before each experiment. The heat source consisted of an iron block heated to varying temperatures and positioned at a distance of 2 cm below the sample.

### Light excitation

We employed a continuous laser (532 nm, 2 W, ROITHNER) for light excitation of red-colored LCE, and a continuous laser (635 nm, 1 W, ROITHNER) for excitation of green-colored LCE. Both laser beams were focused using a plano-convex lens (100 mm, THORLABS) to achieve a focal spot approximately 100 microns in diameter near the baffle edge of the operator. Additionally, an LED source (460 nm module, CoolLED pE-4000) was utilized for supplementary photothermal deformation.

**Tracking method**

Canon 5D Mark III camera with a 100 mm lens and an iPhone 15 smartphone were used to capture optical images and videos. Kinovea program was used to track the position of the baffle. Positional tracking was performed simultaneously for all oscillators. A coordinate offset exists between samples, and its value depends on the experimental conditions.

**Nonlinear dynamics modeling**

See details in Supplementary Notes.

## Data availability

The main data generated in this study are provided in the article and Supplementary Information. The raw data generated in this study have been deposited in the Fairdata database under CC BY 4.0 License, https://doi.org/10.23729/fd-bf56d25e-cef7-394d-a92e-719a4a57d855.

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

## Acknowledgements

The authors gratefully acknowledge funding from Academy of Finland (postdoctoral researcher No. 347201, research fellow No. 340263 and PREIN, No. 320165). H.Z. is thankful to the financial support of the European Research Council (Starting Grant project ONLINE, No. 101076207). H. Z. thanks Hang Zhang (Aalto) for the discussion. A heartfelt thank you to Haotian Pi from Aalto University for the inspiring discussion on the topic of Dissipative Structures. We thank Tampere University for its generous support, and for the unlimited supply of coffee and tea that fueled our manuscript revisions.

## Author contributions

H.Z. conceived the idea and supervised the project; H.G. performed experiments with the help of J.Y., F.L. and H.Z.; D.L. and H.G. performed the electric control experiments. H.G., H.Z. and K.L. analyzed the data; K.L. performed the modeling and developed the concept. H.Z. wrote manuscript with the helps from others; All the authors discussed and contributed to the project.

## Competing interests

The authors declare no competing interests.
