## [Transparent Peer Review file · Nature Communications]

Light-mediated communication in responsive materials ranging from individual self-oscillators to feedback-driven network

Corresponding Author: Professor Hao Zeng

Version 0:

Reviewer comments:

Reviewer #1

(Remarks to the Author)

This manuscript by Zeng et al. describes a design strategy for constructing biomimetic communicative systems by coupling two (or more) oscillators via optical feedback in light responsive LCE materials. Instead of using the conventional self-shadowing effect, this work utilizes a baffle and a soft LCE actuator to generate the required negative or positive feedback loop critical for the creation of light-mediated interindividual communication behaviors. Based on this principle, the authors develop multiple communicative systems consisting of two, three and four oscillators and demonstrate examples of light-mediated communication utilizing commonplace materials, including a thermometer spring and paraffin. Generally, the communicative function in response to light is interesting, and the concept presented in this work is well illustrated. However, I have some concerns and specific questions on this work before I could imagine it would be suitable for publication at Nat Commun.

First, there is no further design or innovation in the LCE material itself, as it has been reported and widely used in prior research. In contrast to its natural counterpart, which is capable of self-adapting to external signals, the communicative behavior observed in this work does not originate from self-adaptation of the material itself but primarily requires a strictly precise setup and highly depends on specific irradiation positions and angles.

Second, the photomechanical actuation mechanism of LCE strip is not clearly described in the current manuscript, is it photochemical or photothermal driven? What is the bending direction, either toward or away from light source? In addition, how the laser was applied to the LCN operator is also very confusing described in the manuscript, especially in the supporting movies.

The specific questions are listed below:

1. In Fig 1c, the definition of the (-) oscillator and (+) oscillator are not illustrated clearly. Adding arrows to indicate the laser direction might be helpful.
2. The authors used red and/or green lasers in different experiments. How do the authors choose a specific one and in which case they need these two?
3. The light intensity unit used in the article in Figure 1 should be mW/cm² instead of mW.
4. In Fig S3, why does the oscillator bend when the whole temperature rises? A clarification needs to be added.

Reviewer #2

(Remarks to the Author)

This manuscript by H. Guo et al. reports on the synchronized movement of two photo-responsive polymers via feedback loops. The polymers are LCE actuated by a photo-thermal effect and oscillate continuously under light irradiation by a self-shadowing effect. By precisely positioning the polymer cantilevers relative to each other with respect to the laser beams, synchronized oscillations of two, three, and even four cantilevers are achieved. Theoretical modeling of the motion is investigated.

The systems developed are ingenious and provide eye-catching experiments to show synchronized movements of plastics triggered by light. By cleverly engineering the system, the authors show how changes in the motion of one oscillator can influence the motion of another. The work is based on photo-actuators, which are well described in the literature and widely

used. And unfortunately, the results do not bring new fundamental insights to both fundamental science and technology, as the photo-induced movement of LCE cantilevers by light reflection has already been reported in Nature Communications (Nature Communications volume 8, article number: 15546 (2017)). Furthermore, the modeling results are qualitatively consistent with the experimental observations, and it is not clear why they are not quantitatively described. I think this work is another demonstration of what can be achieved with LCE actuators, and I do not think it is novel enough to justify publication in Nature Communication. Therefore I recommend that the work be considered for possible publication in a more specialized journal.

Version 1:

Reviewer comments:

Reviewer #1

(Remarks to the Author)

First, I appreciate that the revised manuscript has addressed my previous specific technical issues by providing additional experimental results. Second, however, my concern regarding the lack of further innovation in the materials themselves still remains. After reading the paper again, I still have a feeling that the observation of adaptive behaviors looks highly relies on external environmental factors rather than just responding to the multiple deformation modes of the material itself. Although the author's explanation of the terminology of adaptation makes sense to me, the communicative behavior observed in this work primarily requires a strictly precise setup and highly depends on specific irradiation positions and angles. The authors claim that they outline a general research pathway for communication among active materials, however, whether similar behavior could be reproduced in other materials subject to the strict setup and requirement of irradiation conditions obviously remains debated.

Reviewer #5

(Remarks to the Author)

To the authors,

I have read the paper by Zeng and co-workers with great interest! Their work describes the photomechanical control of self-oscillating elastomers via an optical feedback loop. These self-oscillating elastomers, based on liquid crystalline elastomers (LCEs), utilize a baffle to regulate the optical feedback.

The manuscript is well-written and clearly highlights the importance of using light as a tool for communication between individuals. The figures effectively illustrate this concept. While the use of light-responsive materials such as LCEs for self-oscillation via self-shadowing triggered by actuation is not new, the novelty of this study lies in its design setup. Specifically, it leverages optically controlled self-oscillating behavior in LCEs by integrating a baffle and LCE within a feedback loop, which is crucial for enabling light-mediated communication between individuals.

I believe this manuscript should be published; however, I do not think it is suitable for Nature Communications, as both the self-oscillation concept and the material design of the LCEs are not fundamentally novel. Therefore, I suggest that this work be considered for publication in a more specialized journal.

Here are some suggestions for the authors to further improve their manuscript:

Major reviews.

1- The self-oscillation is clearly induced by photothermal effects. It would be beneficial to include basic material characterization of the LCE, such as the nematic-to-isotropic transition temperature, modulus, and other parameters, and to evaluate whether these material properties influence the self-oscillation behaviour.

2- Is there a way to measure or control the amount of dye (Dispersed Red 1 and Dispersed Blue 14) that has been diffused into the LCE?

3- Figure 1 is critical for explaining the core concept of the study. However, its current format is cluttered and challenging to follow. To enhance clarity, I recommend splitting it into two separate figures: Figure 1: Focus on the mechanisms (e.g., a-f). Figure 2: Highlight the experimental results and key data (e.g., h to g). This separation will improve readability while maintaining the logical flow of the concept.

Reviewer #6

(Remarks to the Author)

The manuscript titled "Light communicative materials" proposes the use of light to couple photothermally deformable objects such as LCE films. An opaque, non-responsive object is attached to the of a photo-responsive object to form a basic unit that is defined as operator. Experimental and numerical results are presented for one self-coupled operator and for two mutually coupled operators. A self-coupled operator exhibits self-oscillation and two mutually coupled oscillations exhibit coupled oscillatory dynamics. Further perturbations (such as wind and light) are applied to one of the mutually coupled operators, and such perturbations lead to the disturbance of the coupled dynamics of the second operator. The coupling is experimentally demonstrated for various photothermally deformable objects and the proposed light-based coupling is further

exploited to demonstrate adaptation, control and sensing applications of the coupled operators. In simple terms, this manuscript proposes a method to replace physical springs with a laser in coupling objects.

Although the proposed concept seems interesting and the demonstrations are impressive, the authors seem to overclaim the significance of the work and the comparison to biological systems seem rather superficial. Additionally, there are many inconsistencies among the experimental data for the same scenario and the proposed analytical model is not fully convincing (see detailed comments below). The manuscript is very cumbersome to read. A thorough and careful rewriting of the manuscript will be helpful in conveying the message concisely. With the above comments in mind, the experiments and the proposed concept of light-based coupling is interesting and offers some advantage over mechanical coupling of objects (such as longer range and smaller delays). Therefore, the paper may be considered for publication after the following comments are addressed:

- 1) In line 48, what do authors mean by mutual interaction mainly through contact-based methods? There are known physical interactions that are not contact-based, such as hydrodynamic interactions and magnetic interactions.
- 2) The claim in lines 56-57 (A contactless approach...emulating biological systems...) seems superficial and unsupported. Why is low temporal delay required to emulate biological communication at cell-level (the reference cited for this line is on cell biology).
- 3) The setting (and definition) in Fig. 1d seems opposite to that shown in supplementary fig. 4.
- 4) Why is the comparison of model and experiments not shown for a single self-oscillator in fig. 1e?
- 5) In Fig. 1g, why is the deformation of -ve operator always positive but +ve operator is oscillating symmetrically about 0?
- 6) Inconsistencies in the data:
 - a. The data in supplementary Fig. 2.3 does not match with that shown in Fig. 1e. In the supp. fig, the amplitude saturates around 1.5 mm after 500 mW, but in Fig. 1e, the amplitude keeps increasing above 1.5 mm, even at 1000 mw.
 - b. plot in Fig. 1g is inconsistent with results in supp fig. 4d (bottom) and supp fig. 6c. The displacement curves don't match. Why is the data so different (period is 1 s in fig. 1 but around 1.7 s in supp fig. 4d)? The amplitudes and period are higher than the theoretical prediction and they don't fit the predicted trend. The data in supplementary fig. 6c does not seem sinusoidal.
 - c. The amplitude for -ve operator is around 1 mm but for the same laser intensity in Fig. 1h, the amplitude is around 4-5 mm. Why?
 - d. Colour coding of the data in Fig. 1g can be made consistent with that of the images in Fig. 1f for better clarity.
 - e. In lines 137-138, text refers to green and red LCEs in Fig. 1c, but the Fig. shows blue and red LCEs.
 - f. Why is the experimental data in supp. fig. 8c inconsistent with the data in Fig. 1h-i?
 - g. Why does the data in suppl. fig. 23b not match with the data shown in Fig. 3e even though they are from the same experiment?
- 7) Does the phase difference/time delay between two coupled operators depend on the size of the baffle and thus the minimum deformation of the LCE required to block/unblock the laser spot?
- 8) In suppl. fig.3, the curvature is defined as 1/radius. The radius cannot be negative, then how is curvature negative?
- 9) Why is the data in suppl. fig.3 presented for multiple trials on only 1 sample? How much does the statistics vary over different samples? and would such variation affect the main results of the paper?
- 10) In line 117, what is the threshold for laser power needed? what factors does this threshold depend on?
- 11) In suppl. fig. 10, why does the - operator not revert back to its original oscillation curve? There is a positive offset in the curve after the disturbance is removed.
- 12) How robust are the oscillator dynamics to external perturbations? Does the dynamics/coupling change after multiple perturbations, or after a strong disturbance?
- 13) The coupling strength/communicated information is not quantified for any case. Such quantifications may be helpful in better understanding the strength of the work.
- 14) There are several concerns regarding the model:
 - a. The only direct dependence on the laser power in the model appears for $A=a(1-\exp(-P/P_0))$. Given this assumption, it is not surprising at all why the deformations follow the same curve suppl. Fig. 2.3. Can you justify why this assumption is made? What is the physical motivation for such assumption? Does the data in Fig. 3b support this assumption?
 - b. In the model, the baffle size is not considered (baffle is implicitly treated as a point object). Therefore, why does the operator require a critical deflection in the simulations to result in oscillation? Why is such critical deflection needed in the experiments?
 - c. Why is the force from laser power modelled as a spring force? What is the physical motivation for the choice? Does a single beam oscillate on constant illumination? It does not appear to be the case.
 - d. How realistic are the chosen simulation parameters? For example, is the choice of $P_0 = 2-3$ W physically meaningful? Please report the values of all the parameters of the model that were used for all the calculations.
 - e. It is not clear what causes the slow dynamics of the coupled oscillators compared to the single oscillator. It would be helpful to add a simple physically meaningful explanation. Lines 147-148 are not clear and seem superficial. In the model this delay is somehow hardcoded by the conditions on displacements.
- 15) The conversion of deformation to electric signal is creative and could be useful. However, In Fig. 5a, why is the shape of the measured signal different from the shape of the deformation, especially because the deformation is linearly related to the incident power (also assumed in the model)?
- 16) The definition of adaptation seems very different from the commonly perceived meaning of adaptation and this definition is not backed up with any references. Therefore, the definition appears to be the opinion of the authors and not an established fact. The results on "adaptation" do not seem to add much value to the paper. Many known materials, for example, shape memory alloy, clay, bistable beams, all seem to fit this definition of adaptation. The results of Fig. 4 could be taken out and reported separately after thorough and careful analysis. This reduction will also help to make the paper more concise and easier to read.

17) The figures appear pixelated and images are unclear. Improving the resolution of the images would help in better appreciating the figures.

Version 2:

Reviewer comments:

Reviewer #1

(Remarks to the Author)

I have followed this manuscript from its first-round submission to Nature Communications and also reviewed critiques and comments from non-listed reviewers in prior rounds of review. Based on the overall comments from the reviewers and the corresponding response from the authors after two-round revisions, I would like to point out that the authors have accepted the criticism regarding material's limited novelty, conclusion overclaim, confusing logic flow, among others, and they revised the manuscript accordingly. This can be seen from the title "Communicative materials" change to a more precise specific one, the adding of a dedicated "system limitations" section to differentiate this work from prior literature, and the extension of the "discussion and conclusions" section. These revisions reflected the authors' positive attitude to improve the quality of the paper, which is good and obviously should be appreciated. However, the acceptance of these criticism raises another fundamental question: Does the revised work meet Nature Communications' threshold for novelty? As a chemist/materials scientist, I recognize that while no new materials were designed, the feedback-driven network oscillation mechanism presents a compelling interdisciplinary contribution, which holds interest for materials science, engineering, and bionics communities. Therefore, I recommend a publication opportunity pending consensus among other reviewers.

Reviewer #5

(Remarks to the Author)

The authors have adequately addressed all of my questions and concerns. I therefore recommend this manuscript for publication in Nature Communications.

Reviewer #6

(Remarks to the Author)

I appreciate the revisions made to the manuscript in response to my previous comments. The quality and clarity of the manuscript seems to be improved. However, some concerns regarding "contact-based interactions", "adaptation", robustness to external perturbations, and generality and scalability of this work remain and it is important that these concerns are addressed via concise discussions and rewriting to clarify the real significance of this work to a broader research community and to avoid largely underestimating the existing literature.

Contact-based interactions: The authors consider hydrodynamic interactions as contact-based, however these interactions do not require a direct physical contact between two interacting objects, only the contact between an object and its surrounding fluid is necessary. The way authors define "contactless methods" seems to eliminate every possible interaction mechanism and only leave interactions mediated through electric, magnetic fields and electromagnetic waves. Even in those, the authors eliminate the field-mediated interactions. The argument regarding the exclusion of magnetic interactions is weak. These interactions do not always need the existence of external fields, for example, magnetic dipole-dipole interactions can exist in the absence of external fields. Besides, a "constant light beam" is not static itself, light is oscillating electric and magnetic fields. Finally, a constant electric field can actually induce oscillatory behaviour of a solid object within a fluid (see the literature on Quincke rollers). The newly added statement regarding magnetic field driven microrobots is incorrect because the magnetically driven microrobots (including those in ref 30) utilize hydrodynamic interactions and magnetic dipole-dipole interactions between the microrobots and only magnetic driving is not enough. Robots in another reference on microrobot swarm cited by the authors (Ref. 29) actually use light-based (IR light) methods for local interactions.

If the authors intend to limit the discussion to light (and the work does seem focused on self-oscillators), then I suggest that they simply motivate the need for light-based interactions and its significance and leave out the discussion on so called "contact-based" interactions. The significance of light-based interactions and its advantages over other interaction mechanisms (if any) needs to be clarified (perhaps related to its low temporal delays and long-range) as it is not entirely clear in the current state of the manuscript. In my opinion, the current text wrongly undermines the literature on other interaction mechanisms. Therefore, the motivation of this work really needs to be strengthened to convey the significance of using light-based interactions for a broader research community (and not just the community on light-responsive self-oscillators), without wrongly undermining the existing literature on other interactions mechanisms.

Adaptation: I appreciate the response from the authors regarding their view on adaptation, however I still think that adaptation is defined and used here based on the subjective opinion of the authors and not on an established fact. Perhaps authors are confusing adaptation with memory? I see their argument regarding the use of feedback loop to induce memory-like effects, however this effect remains highly specialized/specific, limited and constrained by various experimental factors, for example, in the gripper demonstrations the operators have to be precisely placed with respect to the laser beams and therefore the introduction of the local stimulus has to be very precise too. These limitations must be clearly discussed along with the generality of these results in the context of adaptation. The authors mention that reproducibility in such soft matter systems remains limited and that there is a drift in their dynamics over time, given this comment, is the frequency

“adaptation” experiment reproducible? If not, is this experiment meaningful?

Robustness to external perturbations: Perhaps the authors misunderstood my previous comment on robustness of oscillators dynamics. I can see from the figures that the oscillators return (almost always) to the original state after the external perturbation is removed. However, my question was related to the robustness in terms of amount of such experiments that can be performed before the oscillator dynamics is significantly changed and the coupling is lost (if at all). The reported experiments contain only one perturbation cycle and are run for a maximum of 5-6 mins at best. Do the oscillator dynamics and most importantly the light-based interactions persist after long time and after repeated perturbations?

Generality and scalability: The demonstration of light-based coupling in other materials is impressive, however the claim that this strategy does not rely on specific material property seems too broad and unfair and should be alleviated because the coupling is shown for a few soft and light-responsive materials only. Regarding the scalability of this approach, Can the light-based coupling be extended to many operators? Can many operators simultaneously interact with each other (still in pairwise fashion)? Do you need a laser for every operator? If yes, then how does this affect the scalability of this approach to a relatively large network?

Version 3:

Reviewer comments:

Reviewer #6

(Remarks to the Author)

I thank the authors for addressing my concerns with an open mind. The authors have addressed all my concerns and I recommend the manuscript for publication.

Re: Light communicative materials (manuscript NCOMMS-24-40904-T), by H. Guo, et al.

Point-by-point responses to the Reviewers

Reviewer #1: *This manuscript by Zeng et al. describes a design strategy for constructing biomimetic communicative systems by coupling two (or more) oscillators via optical feedback in light responsive LCE materials. Instead of using the conventional self-shadowing effect, this work utilizes a baffle and a soft LCE actuator to generate the required negative or positive feedback loop critical for the creation of light-mediated interindividual communication behaviors. Based on this principle, the authors develop multiple communicative systems consisting of two, three and four oscillators and demonstrate examples of light-mediated communication utilizing commonplace materials, including a thermometer spring and paraffin. Generally, the communicative function in response to light is interesting, and the concept presented in this work is well illustrated. However, I have some concerns and specific questions on this work before I could imagine it would be suitable for publication at Nat Commun.*

Our answer: We are grateful for the reviewer's positive comments. Please see below our point-by-point response to the reviewer's concerns.

-- First, there is no further design or innovation in the LCE material itself, as it has been reported and widely used in prior research. In contrast to its natural counterpart, which is capable of self-adapting to external signals, the communicative behavior observed in this work does not originate from self-adaptation of the material itself but primarily requires a strictly precise setup and highly depends on specific irradiation positions and angles.

Our answer: Thank you for raising this important question. We agree with the reviewer that there is no innovative design in the LCE material itself. However, we would like to point out that the novelty of the manuscript lies in the design of self-oscillating structures and their optical communication pathways. Thus, the implementation of a new LCE material is not the focus of this study. Additionally, we would like to highlight that this study provides a general design framework for various types of responsive materials to achieve light communication, as demonstrated using everyday materials such as thermometers and paraffin.

About the adaptation: In our opinion, the term "adaptation" has been widely misused in the literature. In most case studies, it is used to illustrate the concept of stimulus-responsiveness rather than true adaptation from a bioinspiration perspective. Our viewpoint on adaptation in responsive materials is that a material possesses a state (such as shape or frequency), which can change in response to external stimuli and maintain the altered state even after the stimuli cease. We acknowledge that we previously overlooked the illustration of this important concept. In the revised manuscript, we have included illustrations to reflect three key bioinspired adaptation concepts: bi-stability, cascading transition, and dual rhythms. These now read as follows:

Responsive materials can self-adapt to external stimuli, and the concept of adaptation has been demonstrated in various forms. Here, we explore the potential of using optical feedback loops in communicative materials to program three distinct types of adaptivity: bi-stability, cascading transitions, and dual rhythms.

Figure 4. Adaptation enabled by light-communicative network. (a) Working principle of conventional stimuli-responsive materials. (b) Concept of material state induced by a positive feedback loop, where adaptation refers to the change in state in response to external stimuli. (c) Top: Photographs of an operator-based gripper initially in an open state, adapting to a hot object and changing to a closed state. Bottom: Displacement data of the actuator tip, D_{tip} , in response to heat from an $80\text{ }^{\circ}\text{C}$ object. Insets: Thermal camera images of the system before and after cooling to room temperature. (d) Top: Photographs of an operator-based gripper initially in a closed state, adapting to a cold object and opening. Bottom: Displacement data of D_{tip} in response to cold stimulus from a $0\text{ }^{\circ}\text{C}$ object. Insets: Thermal camera images of the system before and after warming to room temperature. (e) Schematic diagram of the optical pathway in a network of 5 positive operators, showing two stable states (all-open and all-closed). (f) Snapshots of five operators responding to a mechanical trigger on the first operator. Lower right: Change in baffle position (Δd) for each operator. (g) Bifurcation diagram, in which the state change of each operator is recorded as different trigger intervals are applied to the first operator. (h) Design of a dual rhythmic self-oscillator based on two feedback loops. (i) Schematic of the feedback mechanism in the dual rhythmic system. I_1 , intensity of beam 1. D_1 , displacement of operator 1. D_2 , displacement of operator 2. (j) Snapshots of the operator 1 oscillating at different frequencies. (k) Displacement data of the oscillator. Laser used in (c, d): 532 nm, 220 mW, 2 mm. Laser used in (e - g): 532 nm, 100 mW, 4 mm. (h-k) Beam 1 excited on operator 1: 532 nm, 180 mW, 2 mm; beam 2 spot on operator 2: 532 nm, 180 mW, 2 mm; beam 3 spot on operator 1: 532 nm, 160 mW, 2 mm. All scale bars are 1 cm.

Conventionally, stimuli-responsive materials can change their shapes, as illustrated in Figure 4a. For example, a thermally sensitive polymer strip bends when heated and returns to its original, unbent state after removing the heat. Here, we define "adaptation" as the ability of a material to change its state in response to external stimuli and maintain the altered state even after the stimulus is removed. The material's deformation is not directly powered by the stimulus but rather operates as a self-powered, multi-stable system that can switch between stable states upon external disturbances. In other words, it is the bifurcation of the dissipative system that enables dynamical change of condition by self-organizing into different states. Figure 4b demonstrates an example of thermo-adaptation, which differs from the conventional principle of stimuli-responsiveness by emphasizing the material's ability to maintain its state after the stimulus is turned off. To achieve bi-stable states, we design the system with a positive feedback loop, as positive feedback enables an "all-in" or "all-out" bistable mechanism. Supplementary Fig. 27 illustrates this operator and the surrounding optical excitation pathways, where the LCE strip serves as a proof-of-concept gripping device. In this example, a gripper closes when the LCE strip bends and reopens when the strip unbends. Figure 4c depicts a scenario where the gripper, initially open (with a baffle blocking the light beam so the LCE is inactive), closes when exposed to a heated object (80 °C). The bending of the structure unblocks the light beam, delivering energy to the actuator within the positive feedback loop. As a result, the gripper remains closed even after the object's temperature cools to room temperature. Note that there are two stable states of a positive feedback-driven system, as they can be mutually switched by mechanical triggers. Figure 4d presents the opposite case, where the gripper starts in a closed state (with the baffle unblocking the light beam, bending the LCE strip). When exposed to a cold object (0 °C), the gripper opens, and the baffle blocks the light beam. After the object warms to room temperature, the light beam remains blocked, and the gripper maintains its open state.

A series of positive operators connected in a closed feedback loop can result in a cascading transition of material states. This means that switching the state of one operator can trigger a sequential shape-change in others until all operators complete their transition. Figure 4e illustrates an example of five positive operators coupled by five laser beams in a closed loop. The system exhibits two stable states: (1) all operators are open, allowing all the light beams to pass through, and (2) all operators are closed, blocking all beams. Figure 4f captures snapshots of the transition, starting with all operators in the open state. A mechanical trigger is applied to the first operator to switch it OFF, which sequentially drives the 2nd to 5th operators into the closed state (Supplementary Movie 8). Displacement data shown in the lower right corner of Figure 4f highlights the kinetics of the shape-morphing process, revealing a time delay between the transitions of sequential operators. For more detailed displacement data during the cascading transition, see Supplementary Fig. 28. State (shape) transition can be seen as bifurcation of the system, which depends on the strength of the external stimulus, particularly the triggering period. Figure 4g shows that a triggering period of less than 13 seconds cannot induce a complete transition. In such cases, the recently switched operator reverts to its previous state, reversing the sequence. However, for a triggering period of 13 seconds or more, all operators switch to the closed state, and the system stabilizes in its new configuration. In this case, bifurcations signify a change in the network's morphology, characterized by the shape-switching along operators. The bifurcation diagram of the all-closed to the all-open state transition is also shown in Supplementary Fig. 29.

The examples above illustrate the concept of adaptation in bi-stable states, demonstrated through shape changes in a single operator and sequential shape morphing in spatially separated operators. Oscillation frequency serves as another physical metric for assessing the state's condition. In the following, we demonstrate that bifurcations signify a change in the system's behavior, *i.e.*, transitions between oscillatory rhythms.

Figure 4h presents the optical design of a material system incorporating two negative feedback loops. A negative feedback loop induces homeostasis-like self-oscillation with a well-defined frequency (see Figure 1e). The baffle on operator 1 contains two holes that act as distinct negative operators, controlling the passage of beam 1 & 2. Beam 1 is directly reflected back to operator 1, inducing a self-oscillating system in accordance with the principles illustrated in Figures 1d & 1e. Beam 2 interacts with another negative operator, which modulates beam 3. Beam 3, in turn, excites operator 1, following the coupled oscillator mechanism outlined in Figures 1f–1i. Each feedback loop includes a single negative operator and produces a stable self-oscillation frequency. A mechanical trigger allows switching between two oscillation rhythms, as schematically depicted in Figure 4i and Supplementary Movie 9. Governed by the first feedback loop, the system self-oscillates at a high frequency (5.5 Hz), as shown by the photograph and displacement data in Figure 4j. After switching to the second feedback loop, which couples two operators, the system exhibits a lower frequency due to the increased delay within the loop (see Eq. 2 and Figure 1i). The displacement data showing different frequencies and transitions between rhythms are provided in Figure 4k and Supplementary Fig. 30. In summary, the examples above illustrate the adaptation concept enabled by optical feedback loops in responsive materials. We demonstrated self-adaptation through shape changes and rhythm switching in response to external stimuli, as well as the system's capacity to maintain these altered states (shape or rhythm) after the stimulus is removed.

## Supplementary Figures.

**Supplementary Figure 27. Bi-stability enabled by positive feedback.** (a) Optical design of a positive feedback loop consisting of one positive operator and a single light beam. Photos of the operator in the (b) open state and (c) closed state. (d) Four scenarios illustrating thermal adaptation, showing shape changes in response to heat or cold stimuli, and the capability of maintaining the state after ceasing the stimulus. Scale bar is 1 cm.

Supplementary Figure 28. Cascading events in state transition. The network consists of five positive operators in a closed loop, initially all in the closed state. The state changes of each operator are recorded as varying trigger intervals are applied to the first operator.

Supplementary Figure 29. Displacement data during cascading transition. (a-c) Initially, all operators are in the open state. The change in baffle position (Δd) for each operator is recorded by applying a mechanical trigger on operator 1 for (a) 2.5 s, (b) 5 s, and (c) 13 s. (d-f) Initially, all operators are in the closed state. The change in baffle position (Δd) for each operator is recorded by applying a mechanical trigger on operator 1 for (a) 2 s, (b) 4 s, and (c) 7 s.

Supplementary Figure 30. Change in rhythm. (a) Oscillation data showing the transition from low-frequency to high-frequency oscillation triggered by a mechanical stimulus. (b) Oscillation data illustrating transitions from low to high frequency and back to low frequency, induced by different mechanical triggers.

-- Second, the photomechanical actuation mechanism of LCE strip is not clearly described in the current manuscript, is it photochemical or photothermal driven? What is the bending direction, either toward or away from light source? In addition, how the laser was applied to the LCN operator is also very confusing described in the manuscript, especially in the supporting movies.

Our answer: Thank you for your question. The actuation mechanism operates based on the photothermal effect. We have provided an explanation of the photo-actuation mechanism. Here is the updated description

The sample was prepared with a splayed alignment, resulting in bending deformation upon thermal stimulation. When exposed to light, the dyes within the LCE convert photon energy

into heat, causing the strip to bend toward the contracting surface (planar alignment), regardless of the direction of incidence.

4. Supplementary Video captions

Supplementary Movie 1. Coupled self-oscillators.

This real-time video shows continuous oscillations of two coupled operators. The black markings facilitate position tracking during the oscillation. Laser 1 (right to left): 532 nm, 150 mW, 2 mm spot size. Laser 2 (left to right): 635 nm, 200 mW, 3 mm spot size. LCE actuator dimensions: $24 \times 2 \times 0.1 \text{ mm}^3$. Baffle dimension: $5 \times 20 \times 0.01 \text{ mm}^2$. The optical pathway follows the same design in Fig. 1f.

Supplementary Movie 2. Coupling between two isolated self-oscillators.

This real-time video shows continuous oscillations of two coupled operators with a screen in the middle. The laser beams are guided around the screen and reflected onto the sample through mirrors. Laser 1 (exciting on left sample): 532 nm, 150 mW, 2 mm spot size. Laser 2 (exciting on right sample): 635 nm, 300 mW, 3 mm spot size. LCE actuator dimensions: $24 \times 2 \times 0.1 \text{ mm}^3$. Baffle dimension: $5 \times 20 \times 0.01 \text{ mm}^2$. The optical pathway follows the same design in Fig. 2a.

Supplementary Movie 3. Coupling between two self-oscillators through optical fibers.

This real-time video shows continuous oscillations of two coupled operators through optical fibers. Laser 1 (exciting on upper sample): 532 nm, 256 mW, 3 mm spot size. Laser 2 (exciting on bottom sample): 532 nm, 200 mW, 2 mm spot size. LCE actuator dimensions: $24 \times 2 \times 0.1 \text{ mm}^3$. Baffle dimension: $5 \times 20 \times 0.01 \text{ mm}^2$. The samples and connections to fibers are shown in Fig. 2h.

Supplementary Movie 4. Coupled network composed of three units.

This video shows continuous oscillations of three coupled operators. The laser beams are guided through optical fibers. All laser beams: 532 nm, 320 mW, 2 mm spot size at the sample positions. LCE actuator dimensions: $24 \times 2 \times 0.1 \text{ mm}^3$. Baffle dimension: $5 \times 20 \times 0.01 \text{ mm}^2$. The movie is played with 4× accelerated speed. The optical pathway is schematically shown in Supplementary Figure 18c.

Supplementary Movie 5. Coupled network composed of four units.

This video shows continuous oscillations of four coupled operators. The laser beams are guided through optical fibers. All laser beams: 532 nm, 320 mW, 2 mm spot size at the sample positions. LCE actuator dimensions: $24 \times 2 \times 0.1 \text{ mm}^3$. Baffle dimension: $5 \times 20 \times 0.01 \text{ mm}^2$. A red coloured filter is used to block laser wavelengths for video recording. The movie is played with 4× accelerated speed. The optical pathway is schematically shown in Supplementary Fig. 18b.

Supplementary Movie 6. The coupling between a thermometer and LCE actuator.

This video shows continuous oscillations of the coupled system. Laser 1 (exciting on LCE): 532 nm, 44 mW, 2 mm spot size. Laser 2 (exciting on thermometer): 532 nm, 930 mW, 3 mm spot size. LCE actuator dimensions: $24 \times 2 \times 0.1 \text{ mm}^3$. Baffle dimension: $5 \times 20 \times 0.01 \text{ mm}^2$. The movie is played with 2× accelerated speed. The optical pathway is schematically shown in Supplementary Fig. 24c.

Supplementary Movie 7. The coupling between the paraffin and LCE actuator.

This video shows continuous oscillations of the coupled system. Laser 1 (exciting on LCE): 532 nm, 80 mW, 1.2 mm spot size. Laser 2 (exciting on paraffin): 532 nm, 1440 mW, 2 mm spot size. LCE actuator dimensions: $24 \times 2 \times 0.1 \text{ mm}^3$. Baffle dimension: $5 \times 20 \times 0.01 \text{ mm}^2$. The movie is played with 16× accelerated speed. The optical pathway follows the same design in Supplementary Fig. 26.

Supplementary Movie 8. Cascading transition.

This real-time video shows the cascading transition from ON to OFF state and then from OFF to ON state. All Laser: 532 nm, 100 mW, 2 mm spot size. LCE actuator dimensions: $16 \times 2 \times 0.1 \text{ mm}^3$. Baffle dimension: $4 \times 12 \times 0.01 \text{ mm}^2$. The optical pathway follows the same design in Fig. 4e.

Supplementary Movie 9. Dual rhythm in self-oscillation.

This real-time video shows the mechanical trigger induced transition between two oscillation frequencies. Laser 1 excited on operator 1: 532 nm, 180 mW, 2 mm; Laser 2 spot on operator 2: 532 nm, 180 mW, 2 mm; Laser 3 spot on operator 1: 532 nm, 160 mW, 2 mm. LCE fiber dimensions: 1.6 cm in length and 1 mm in diameter. Baffle dimension: $4 \times 12 \times 0.01 \text{ mm}^2$. The optical pathway follows the same design in Fig. 4h.

The specific questions are listed below:

1. In Fig 1c, the definition of the (-) oscillator and (+) oscillator are not illustrated clearly. Adding arrows to indicate the laser direction might be helpful.

Our answer: Thank you for your question and good suggestion. We have revised Figure 1c and added an explanation in the text.

As shown in Figure 1c, when the initial beam is positioned below the baffle edge, the activated LCE moves towards the beam spot, tends to obstruct the light, thus acting as a negative operator. Conversely, when the baffle edge initially blocks the beam, the actuation moves the baffle away from the beam spot, allowing light to propagate, and the system functions as a positive operator.

2. The authors used red and/or green lasers in different experiments. How do the authors choose a specific one and in which case they need these two?

Our answer: We appreciate the feedback. We realized that it was our oversight, and we have corrected it. In the revised version, we have included an explanation. The updated text now reads:

It is worth noting that two colors shown in Figure 1c are used to distinguish between the laser beams and actuators: the red laser activates the green LCE, while the green laser activates the red LCE, providing a clearer illustration of the working principle. The coupling mechanism does not depend on the excitation wavelength. In Figures 2-5, only green laser beams (and red-colored LCEs) will be used for the experiments.

3. The light intensity unit used in the article in Figure 1 should be mW/cm^2 instead of mW .

Our answer: We appreciate the feedback. However, we would like to retain the original description to avoid any confusion. The LCE strip is excited by a laser spot smaller than the strip size, not by global illumination that covers the whole sample area. The former can be experimentally reproduced by specifying the laser power (in mW) and spot size (in mm). The latter is typically described in terms of intensity (in mW/cm^2).

4. In Fig S3, why does the oscillator bend when the whole temperature rises? A clarification needs to be added.

Our answer: Thank you for your positive comment. This is due to the thermally induced deformation in the splayed aligned LCE samples. A detailed revision can be found in the previous response, under the second main point.

Reviewer #2: *This manuscript by H. Guo et al. reports on the synchronized movement of two photo-responsive polymers via feedback loops. The polymers are LCE actuated by a photo-thermal effect and oscillate continuously under light irradiation by a self-shadowing effect. By precisely positioning the polymer cantilevers relative to each other with respect to the laser beams, synchronized oscillations of two, three, and even four cantilevers are achieved. Theoretical modeling of the motion is investigated. The systems developed are ingenious and provide eye-catching experiments to show synchronized movements of plastics triggered by light. By cleverly engineering the system, the authors show how changes in the motion of one oscillator can influence the motion of another.*

Our answer: We are grateful for the comments.

The work is based on photo-actuators, which are well described in the literature and widely used. And unfortunately, the results do not bring new fundamental insights to both fundamental science and technology, as the photo-induced movement of LCE cantilevers by light reflection has already been reported in Nature Communications (Nature Communications volume 8, article number: 15546 (2017)).

Our answer: Thank you for raising this important question. We agree with the reviewer that there is no innovative design in the LCE material itself. However, we would like to point out that the novelty of the manuscript lies in the design of self-oscillating structures and their optical communication pathways. Thus, the implementation of a new LCE material is not the focus of this study. Additionally, we would like to highlight that this study provides a general design framework for various types of responsive materials to achieve light communication, as demonstrated using everyday materials such as thermometers and paraffin.

Furthermore, the modeling results are qualitatively consistent with the experimental observations, and it is not clear why they are not quantitatively described.

Our answer: We appreciate the feedback. We noticed that it was our mistake to overlook the nonlinear aspects of the modeling. In the revised manuscript, we have made substantial modifications to the modeling approach to achieve quantitatively matching results. The updated version now reads as follows:

Main text:

The period of the coupled system is derived and estimated to be $T \approx -4\tau_{\text{heat}} \ln\left(1 - \frac{w_0}{\bar{A}}\right)$ (Eq.2), where τ_{heat} is the characteristic time for heat exchange between photothermally-responsive LCE cantilever and environment, $\bar{w}_0 = \frac{w_0}{l}$ is the dimensionless on/off transition critical deflection of LCE cantilevers, and \bar{A} is the dimensionless limit deflection of the LCE cantilevers under longtime illumination.

Supplementary Figures:

Supplementary Figure 6. Modelling fitting. (a) Experimental result oscillation data of the coupled oscillators. Displacement shows the change of tip positions of the two baffles. Laser 1: 138 mW, laser 2: 210 mW. (b) Theoretical prediction. In the simulation, we set $l_- = 2.5$ cm, $l_+ = 2.5$ cm, $w_{0-} = 0.475$ mm, $w_{0+} = 0.35$ mm, $\bar{\beta} = 0.2$, $\tau_{\text{inertial}} = 0.115$ s, $\tau_{\text{heat}} = 0.287$ s, $P_- = 210$ mW, $P_+ = 138$ mW, $a_- = 0.5$, $a_+ = 1.15$, $P_{0-} = 3$ W, $P_{0+} = 3$ W.

Figure 2.2. Numerical results. Time history of the displacement of the mass block, for three different laser powers $P=120$ mW, 200mW, and 280mW. In the simulation, we set $l=2.5$ cm, $w_0=0.5$ mm, $\bar{\beta} = 0.5$, $\tau_{\text{inertial}} = 0.017$ s, $\tau_{\text{heat}} = 0.17$ s, $a = 0.131$, and $P_0=61$ mW.

Figure 2.3. Dependences of amplitude and period of single oscillator on the laser power. In the simulation, we set $l = 2.5$ cm, $w_0 = 1.5$ mm, $\bar{\beta} = 0.2$, $\tau_{\text{inertial}} = 0.013$ s, $\tau_{\text{heat}} = 0.13$ s, $a = 0.099$, and $P_0 = 66$ mW. Laser spot size: 2 mm.

Figure 2.6. Amplitude and period of two coupled oscillators. In the computation, we set $l_- = 2.5$ cm, $l_+ = 2.5$ cm, $w_{0-} = 4.25$ mm, $w_{0+} = 4$ mm, $\bar{\beta} = 0.8$, $\tau_{\text{inertial}} = 0.017$ s, $\tau_{\text{heat}} = 0.34$ s, $P_+ = 180$ mW, $a_- = 3.42$, $a_+ = 1.83$, $P_{0-} = 2$ W, $P_{0+} = 2$ W. Spot size for laser (+), 2 mm, size for laser (-), 3 mm. Dots: experimental results. Solid lines: theoretical predictions.

Figure 2.7. Amplitude and period of two coupled oscillators. In the computation, we set $l_- = 2.5$ cm, $l_+ = 2.5$ cm, $w_{0-} = 1.625$ mm, $w_{0+} = 3.75$ mm, $\bar{\beta} = 0.8$, $\tau_{\text{inertial}} = 0.02$ s, $\tau_{\text{heat}} = 0.4$ s, $P_- = 150$ mW, $a_- = 2.47$, $a_+ = 2.01$, $P_{0-} = 2$ W, $P_{0+} = 2$ W. Spot size for laser (+), 2 mm, size for laser (-), 3 mm. Dots: experimental results. Solid lines: theoretical predictions.

Other changes are kindly referred to the equations highlighted in the Supplementary Modelling.

I think this work is another demonstration of what can be achieved with LCE actuators, and I do not think it is novel enough to justify publication in Nature Communication. Therefore I recommend that the work be considered for possible publication in a more specialized journal.

Our answer: Thank you for your comment. First of all, we recognize that it was our mistake to overlook the application aspect. Therefore, we have added additional experiments to demonstrate the robotic application. The revised section now reads:

Sensing and control.

Figure 5. Remote sensing and robotic control. (a) Top: Schematic illustration of the two-unit network connected via two light beams. Bottom: Mechanism of signal transmission, showing how material deformation is converted into an electrical signal at a remote distance. (b) Displacement data from operator 1 and the corresponding electric signal received near operator 2. (c) Detection of wind disturbance at operator 1 and subsequent signal transition observed at the detector near operator 2. (d) Top: Schematic of the setup for remote deformation control. Bottom: Explanation of the remote-control mechanism. (e) Displacement of operator 1 and transmitted light power at various applied voltages. (f) Displacement of operator 2 in response to voltage changes applied to the electromagnetic coil near operator 1. (g) Modulation of operator 2 via the application of a square wave signal voltage to the electromagnetic coil. E , electric voltage generated by a photodiode detector or voltage applied to an electromagnetic coil. I_1 , intensity of beam 1. D_1 , displacement of operator 1. D_2 , displacement of operator 2. (b, c) Laser 1: 532 nm, 256 mW, 3 mm. Laser 2: 532 nm, 200 mW, 2 mm. LCE sample dimensions: $24 \times 2 \times 0.1 \text{ mm}^3$, baffle size: $5 \times 20 \times 0.01 \text{ mm}^3$. (e-g) Laser: 532 nm, 200 mW, 2 mm.

For robotic applications, the light communication concept provides a method for remote sensing and electrical control of soft actuators over long distances. Figure 5a illustrates that the deformation of operator 1 affects the transmission intensity of the nearby light beam. By using a photodiode detector, the deformation signal from operator 1 is converted into an electrical signal, which is then received at the location of operator 2, as shown by the displacement and voltage data in Figure 5b. This enables remote sensing, such that, for example, a wind disturbance affecting operator 1 can be detected by the variation in the electrical signal at the position of operator 2 (Figure 5c). More detailed remote sensing data, including long- and short-period perturbations from wind and mechanical triggers, can be found in Supplementary Fig. 31.

Conversely, a tethered voltage signal can induce deformation of operator 1 via an electromagnetic coil, while a light beam can independently control operator 2 over a long distance. The working principle is illustrated in Figure 5d, with setup images provided in Supplementary Fig. 32. When a voltage is applied to the electromagnetic coil, the magnetic field generates an attractive force that deforms operator 1, thereby activating the light beam, as shown by the displacement and light power data in Figure 5e. The transmitted beam then excites operator 2, causing deformation. Thus, the voltage signal near operator 1 can remotely control the deformation of operator 2 (Figure 5f). A negative voltage creates a propelling force that deforms operator 1 in the opposite direction, causing the light transmission to close. Control data based on the negative voltage signal is shown in Supplementary Fig. 33. Robotic control can also be achieved by modulating the signal with different bandwidths, as demonstrated in Figure 5g and Supplementary Fig. 34.

Supplementary Figures.

Supplementary Figure 31. Remote sensing. Oscillation data of the coupled oscillators in response to external wind disturbance applied to operator 1 for (a) short and (b) long durations, and the received electrical signal at a position near operator 2. Oscillation data of the coupled oscillators in response to external mechanical disturbance applied to operator 1 for (c) short and (d) long durations, and the received electrical signal at a position near operator 2. Laser 1: 532 nm, 256 mW, 3 mm. Laser 2: 532 nm, 200 mW, 2 mm. LCE sample dimensions: $24 \times 2 \times 0.1 \text{ mm}^3$, baffle size: $5 \times 20 \times 0.01 \text{ mm}^3$. Scale bar: 2 cm.

Supplementary Figure 32. Photograph of the electric control system. (a) Side view showing the light-coupled components and electromagnetic coil. Front view images of the system with (b) the 'cut-ON' state activated by a positive voltage and the 'cut-OFF' state activated by a negative voltage signal. Two magnets are mounted on the baffle of one component near the coil. Scale bar is 1 cm. Magnet: Neodymium 50 magnet, 2mm diameter \times 1mm thick, 23mg weight. The coil is made of copper wire with 1,000 turns.

Supplementary Figure 33. Remote control system. (a) Top: Schematic of the setup for remote deformation control. Bottom: Explanation of the remote-control mechanism. (b) Displacement of operator 1 and transmitted light power at various applied voltages. (c) Displacement of operator 2 in response to voltage changes applied to the electromagnetic coil near operator 1. E , electric voltage applied to the electromagnetic coil. I_1 , intensity of beam 1. D_1 , displacement of operator 1. D_2 , displacement of operator 2. Laser 1: 532 nm, 200 mW, 2 mm. LCE sample dimensions: $24 \times 2 \times 0.1$ mm³, baffle size: $5 \times 20 \times 0.01$ mm³.

Supplementary Figure 34. Remote control data. Modulation of operator 2 via the application of a square wave voltage signal. Voltage: (a) 0 to -5 V, (b) 0 to 3 V. 50% duty cycle. D_2 , displacement of operator 2.

Secondly, respectfully, we disagree with the reviewer on “*this work is another demonstration of what can be achieved with LCE actuators*”. The novelties of this work are as follows and have not been reported in any previous literature.

- (1) A refined definition of photomechanical feedback in a newly designed actuator-baffle assembly, laying the **foundation for establishing positive and/or negative feedback loops** to achieve far-from-equilibrium states in light-responsive materials.
- (2) Coupling between two or more self-oscillators via an optical feedback loop for **adaptation**.
- (3) The light-mediated communication method offers **high-directionality** in network connection, facilitating material interaction over significant distances, contactlessly.
- (4) The feedback loop can selectively engage diverse elements, giving rise to adaptation such as **bistable state, cascading transition and dual rhythm**.
- (5) The principle demonstrated herein exhibits **broad applicability** across various responsive materials.

To further highlight the novelties of the manuscript, we have added the following discussion in the revised text. It now reads:

Research on bioinspired materials has evolved from focusing on static functionalities to embracing increasing levels of dynamic responsiveness and interactive behaviors.^{4-6,31} These advancements emphasize the autonomy and adaptability of biological systems, aiming to introduce new interaction behaviors and functions in synthetic responsive materials. For example, the work presented includes methods for mechanical interaction in synchronization,²³ self-oscillation driven by negative feedback from heat diffusion,^{12,13} and spatial patterns induced by feedback from chemical reactions.⁴⁰ In all these systems, feedback plays a crucial role in maintaining the system far from thermodynamic equilibrium, enabling interaction. However, the predominant methodologies rely on contact-based approaches, such as mechanical interaction²³ and the exchange of reactive substances.⁵⁵ These approaches are limited by temporal delays, short transfer distances, and a lack of directionality. In contrast, the method reported in this study couples two or more self-oscillators through an optical feedback loop, providing high-directionality in network connections and enabling material interaction over long distances with minimal delay. This feedback loop can selectively engage different oscillating elements, allowing for adaptation at the system level. The demonstrated principle also has broad applicability across a range of responsive materials.

Re: Light communicative materials (manuscript NCOMMS-24-40904A), by H. Guo, et al.

Point-by-point responses to the Reviewers

Reviewer #1:

First, I appreciate that the revised manuscript has addressed my previous specific technical issues by providing additional experimental results. Second, however, my concern regarding the lack of further innovation in the materials themselves still remains. After reading the paper again, I still have a feeling that the observation of adaptive behaviors looks highly relies on external environmental factors rather than just responding to the multiple deformation modes of the material itself. Although the author's explanation of the terminology of adaptation makes sense to me, the communicative behavior observed in this work primarily requires a strictly precise setup and highly depends on specific irradiation positions and angles. The authors claim that they outline a general research pathway for communication among active materials, however, whether similar behavior could be reproduced in other materials subject to the strict setup and requirement of irradiation conditions obviously remains debated.

Our answer: We are grateful for the reviewer's comments. Please see below our point-by-point response to the reviewer's concerns.

About material design. It is our initial intention to introduce a method through light beams coupling between two pieces of material to achieve material interaction. This method is not intended to rely on the specific design of the material itself. That has been the reason, we have used the most standard liquid crystalline elastomer (LCE) actuator for demonstration. More importantly, we have extended the use of daily materials, i.e., thermally responsive bilayer and paraffin as shown in Figure 5, to achieve the similar results shown in LCE actuators.

About a “general pathway for communication”. We admit that it is our mistake to overstate the results, which causes unnecessary misleading. We agree with the reviewer that, the “communicative” behaviour requires strictly optical alignment arranged between two actuator (operator) samples. This factor has, indeed, limited the generalization of the working principle to be implemented in other material systems. In the revised version, we have included a comprehensive and self-critical discussion about the limitations of the system. Now, they read,

System limitations.

Unlike thermo-responsive cantilevers made of metallic bilayers, soft actuators composed of LCEs exhibit non-uniform deformation along the strip during dynamic oscillation. Supplementary Fig. 31 presents a series of photographs capturing the shape-morphing behaviour of the soft LCE in the experiments, revealing a wide variety of configurations. These variations arise from differences in sample orientation and the position of the laser excitation spot relative to the strip in different experimental setups. Additionally, the light-heat-induced deformability varies among samples due to the self-assembled nature of liquid crystals – even strips cut from the same film prepared in the same cell can exhibit subtle differences in actuation behaviour (Supplementary Fig. 3b). The viscoelastic properties of the soft material further contribute to a gradual drift in oscillation behavior over extended periods, as observed

in the Supplementary Figs. 4, 10, 14, 16, 17, 29. Moreover, this light-mediated interaction mechanism depends critically on the precise alignment of the laser. Two focused laser spots must be positioned near the operators' baffles to enable the cut-on/off action. Details see in boundary conditions in the model: Eq. 11, 12 and Eq. 19, 20 in Supplementary Note. However, the baffle's critical distance w_0 between the cut-on/cut-off position and the system's equilibrium point, cannot be predetermined prior to the onset of dynamic self-oscillation. These factors collectively result in a variation of the delay time, t_d , in the coupled oscillator. Supplementary Note 2.3 shows theoretical prediction data, demonstrating that changes in w_0 influence the oscillation frequency, amplitude, and waveform.

It is noticed that light-mediated interactions in soft matter systems should not aim for precise control or fine-tuning of oscillatory properties across different samples. Instead, this study highlights the significance of achieving non-equilibrium states, enabling signal transmission and feedback-loop-driven adaptation, which are the key features inspired by biological systems.

About the debated topic. We highly appreciate the comment, “*whether similar behavior could be reproduced in other materials subject to the strict setup and requirement of irradiation conditions obviously remains debated*”. In this study, we have shown a similar “communicative” behaviour using thermal bilayer and paraffin (original Fig. 3, new Fig. 5), thus in our opinion, this method can be, in principle, applied to other materials systems. However, we agree with the reviewer that, the overall method, still, highly relies on optical arrangements, as discussed in the previous response. In the revised version, we have included a discussion about different responsive materials and their potential in using the method in this study. We also highlighted a multi-disciplinary view in the research field. Now, they read,

This light-mediated interaction strategy holds promise for application to other responsive materials.^{20,55} Hydrogels, for instance, typically exhibit a lower critical solution temperature (LCST), enabling light-heat triggered opaqueness above the LCST. Zhang et al. employed this light-induced opaqueness as a negative operator to achieve self-oscillation in a laser-excited gel tube.¹³ Similarly, photothermal bilayers, and humidity-sensitive actuators that can mechanically respond due to light-heat-induced desorption,⁵⁶ undergo reversible deformation and thereby function as positive and negative operators, akin to the LCEs utilized in this study. Additionally, joule-heat triggered LCEs¹⁴ and shape-memory alloys⁵⁷ have been used as positive and negative operators within feedback networks built upon by electrical circuits, whose underlying mechanisms are comparable to the light-mediated approach demonstrated here. In previous literature, the emphasis has often been placed heavily on the material properties necessary to reach non-equilibrium states.^{13,58} In contrast, we believe that stimuli-responsiveness is a widespread characteristic, as illustrated by everyday materials shown in Figure 5. We advocate for greater interdisciplinary dialogue to establish a general design framework for communication in synthetic materials.

About overall writing. We have also made an overall examination of the writing style, to ensure an accurate scientific description, and transparency of the system limitation. The title of the manuscript has also been changed to,

Light-mediated interactions in responsive materials: From individual self-oscillators to feedback-driven network.

We believe that underlying concept, limitation and its potential impacts are clearly communicated without overstating in this revised version.

Reviewer #5:

I have read the paper by Zeng and co-workers with great interest! Their work describes the photomechanical control of self-oscillating elastomers via an optical feedback loop. These self-oscillating elastomers, based on liquid crystalline elastomers (LCEs), utilize a baffle to regulate the optical feedback.

The manuscript is well-written and clearly highlights the importance of using light as a tool for communication between individuals. The figures effectively illustrate this concept. While the use of light-responsive materials such as LCEs for self-oscillation via self-shadowing triggered by actuation is not new, the novelty of this study lies in its design setup. Specifically, it leverages optically controlled self-oscillating behavior in LCEs by integrating a baffle and LCE within a feedback loop, which is crucial for enabling light-mediated communication between individuals.

Our answer: We are grateful for the very positive comments, highlighted by “*The manuscript is well-written and clearly highlights the importance of using light as a tool for communication between individuals.*”

I believe this manuscript should be published; however, I do not think it is suitable for Nature Communications, as both the self-oscillation concept and the material design of the LCEs are not fundamentally novel. Therefore, I suggest that this work be considered for publication in a more specialized journal.

Our answer: We are grateful for the reviewer’s opinions. However, we respectfully disagree with the reviewer about the comment, “*I do not think it is suitable for Nature Communications, as both the self-oscillation concept and the material design of the LCEs are not fundamentally novel*”. The reasons are explicated below.

We kindly note that self-oscillation is only one of the indicators for indicating the non-equilibrium state. The driving mechanism behind this is negative feedback that processed within the material’s stimulus-responsiveness. We agree with the reviewers that in the past decades, there exist an increasing number of publications reporting on the self-shadowing oscillators and self-oscillating robots based on them. But, the key challenge is that, **there still lacks** a method to extend such a self-shadowing oscillator to a complex construct to receive practical functions. More importantly, yet no report discussed how to implement the optical feedback to design and attain a sophisticated level of bio-mimic function. Our findings introduce a novel model – an actuator and a baffle operator that are **not** self-shadowed – that can offer both negative and positive feedback. The results not only show a self-oscillator at non-equilibrium state, but also can (1) enable feedback loop connection among multiple individuals, (2) signal transmission between samples over significant distances, contactlessly, (3) memory effect based on positive feedback and (4) a feedback network showing adaptation in both shape-morphing and frequency. We believe the above achievements go beyond the state-of-the-art concept appearing in literature.

In the revised version, we have added a comparative discussion to the literature. Now, it reads,

The self-shadowing effect has been widely employed in the development of LCE self-oscillators.^{11,46} However, most implementations are limited to individual oscillators. Few studies have demonstrated interactions between self-oscillating strips via mechanical vibrations²³ or hydrodynamic forces,²⁴ leaving significant potential for expanding interactive units. Depending on the coupling strength, a disturbance to one oscillator can have varying effects on the other.⁵³ In this study, we present a method to achieve strong coupling between two oscillators using light beams – such that if one oscillator ceases, the other also stops. This behavior is distinct from that observed in existing self-oscillating systems (see Supplementary Fig. 13). This method also offers designs of positive and negative operators that can be programmed to establish diverse feedback loops among multiple oscillators, offering new alternatives for biomimetic research by emphasizing the central role of feedback in the realization of bioinspired functions.

In previous literature, the emphasis has often been placed heavily on the material properties necessary to reach non-equilibrium states.^{13,58} In contrast, we believe that stimuli-responsiveness is a widespread characteristic, as illustrated by everyday materials shown in Figure 5. We advocate for greater interdisciplinary dialogue to establish a general design framework for communication in synthetic materials.

In previous literature, adaptation has often been described as the responsive shape change of a material in reaction to environmental stimuli. In this study, we propose a broader perspective: adaptation can be viewed as the capacity of a material to change its state in response to external disturbances, coupled with the ability to retain this altered state even after the disturbance is removed. Typically, maintaining an altered state requires mechanisms, e.g., shape memory effects,⁵⁹ plastic deformation (as in deformable clays), or bi-stable mechanical architectures.⁶⁰ Here, in Figure 4b–g, we demonstrate that a positive feedback loop can induce bi-stability without relying on shape memory, microstructural changes, or specific mechanical designs. Moreover, a material state can be characterized not only by its spatial configuration (the shape) but also by its oscillation frequency in the time domain.³⁷ Thus, adaptation may also refer to the ability to shift and sustain different oscillation frequencies following external disturbances, as illustrated in Figure 4h–k.

Here are some suggestions for the authors to further improve their manuscript:

Major reviews.

1- The self-oscillation is clearly induced by photothermal effects. It would be beneficial to include basic material characterization of the LCE, such as the nematic-to-isotropic transition temperature, modulus, and other parameters, and to evaluate whether these material properties influence the self-oscillation behaviour.

Our answer: We thank you for the comments. We agree that the material properties, i.e. Young's modulus, dimensions and thermal responsiveness, can affect the oscillation frequency. In the revised version, we have included material characterization for better understanding of the material properties. Now, they read,

Detailed information regarding the chemical structures and preparation process can be found in Supplementary Figs. 1-2. For thermally and photothermally induced bending, temperature-induced strain, and mechanical properties, see Supplementary Fig. 3.

Supplementary Figure 3. Stimuli-responsiveness of LCE actuator. (a) Left: Photographs showing the LCE strip geometries at room condition and elevated temperature. Strip size: $15 \times 1 \times 0.1 \text{ mm}^3$. The sample is placed on top of a hot plate and covered with a transparent glass window to attain homogeneous temperature distribution. Right: The curvature variation upon increasing the temperature. Curvature is defined as $1/r$, where r is the radius of the strip, as shown in the inset. (b) Left: Photographs displaying the LCE strip geometries upon different illuminating intensities. Strip size: $15 \times 1 \times 0.1 \text{ mm}^3$. Right: The bending angle of three independent LCE actuators upon change of illuminating light intensity I . The strips are cut from the same LCE film. The bending angle (α) is indicated in the inset. Error bars represent s.d. for $n = 3$ measurements. The same sample was measured repeatedly. Scale bars in (a, b) are 1 cm. (c) Left: Polarized microscopy images of an LCE film at 60 and 350 °C. Strain curve of LCE upon elevated temperature. L , the length of the film after deformation, L_0 , the original length of the LCE. (d) Tensile testing of an LCE strip. Inset shows the measurements of Young's modulus (E_Y), fracture strain (ϵ_{\max}), and tensile strength (σ_{\max}).

We would like to note that, one of the findings in this study is that, after being coupled with two laser beams the coupled oscillator's frequency is no longer determined by the resonance frequency of the single oscillator. Clear evidence is shown by comparing the data between a single oscillator (10 Hz) and coupled ones with a similar size (about 1 Hz). The large difference in oscillation behaviour lies in a distance-dependent delay process in the feedback coupled oscillator, which has no direct connection to the resonance of single oscillator properties. We have added a Supplementary Note 2.3 for a better explanation, which includes a discussion of how material properties' influences on the oscillation frequency.

2.3 Comparison between single and coupled oscillators

The oscillation behaviour of the single self-oscillator is dominated by the mechanical resonance, of which frequency is governed by Eq. (16). Taking $\omega_0 = 1/\tau_{\text{inertial}} = \sqrt{3\Pi/ml^3}$, $\Pi = E_Y I$, Π is bending stiffness, E_Y , Young's modulus, I , second moment of area, l , m are the length and mass of the LCE cantilever, respectively. $\bar{\beta} = \beta\tau_{\text{inertial}}/m$ (β , damping coefficient). The oscillation frequency of single self-oscillator can be written as,

$$f_s = \frac{\sqrt{\frac{3E_Y I}{ml^3} \left(1 - \frac{\beta^2 l^3}{12E_Y I m}\right)}}{2\pi} \quad (22)$$

Variation of the material's mechanical properties such as change of Young's modulus (referred to tensile testing in Supplementary Fig. 3d), and geometric parameter, i.e. the second moment of area changes directly the self-oscillation frequency of the single oscillator.

After being coupled with two laser beams the coupled oscillator's frequency is no longer determined by the resonance frequency of single oscillator. The large difference in oscillation behaviour lies in a distance-dependent delay process in the feedback coupled oscillator. The oscillation behaviour is predicted by Eq. (21). Restoring from nondimensionalization treatment by taking $\bar{w}_0 = w_0/l$, $\bar{P} = \frac{A\eta}{kl}P$ (A is the light-driven bending coefficient, η is the energy absorption coefficient, k is the heat transfer coefficient), the oscillation period can be written as,

$$T \approx -4\tau_{\text{heat}} \ln\left(1 - \frac{kw_0}{A\eta P}\right) \quad (23)$$

Changing the actuator material alters a complex combination of parameters, including τ_{heat} , k , A , η . For example, reducing the crosslinking density lowers the nematic-to-isotropic phase transition temperature (T_{ni}) from above 300 °C as observed in this study (Supplementary Fig. 3c), to around 100 °C (Ref. 2), which effectively increases A . Increasing the sample thickness, on the other hand, can reduce A while also changing the thermal capacity and increasing the value of τ_{heat} .

2- Is there a way to measure or control the amount of dye (Dispersed Red 1 and Dispersed Blue 14) that has been diffused into the LCE?

Our answer: Thank you for the comments. We acknowledge that this is an excellent technical question. In the new Supplementary Figure 32, we show that a prolonged dyeing process increases absorbance, which eventually saturates at a value of 2 for extended dyeing durations. We would like to note that photothermal actuation in LCEs depends on the total amount of absorbed light. An absorbance value of 2 corresponds to a transmittance of 0.01, meaning that 99.9% of the incident light is absorbed (neglecting reflectance ~3% for a polymer surface). We have modified the text, now it reads,

This process entailed placing the samples on a hotplate set to 100 °C and uniformly dispersing powders of Disperse Red 1 and Disperse Blue 14 dyes onto the surface of the LCE, following a 5-minute thermal diffusion. The absorbance saturates around 2 after 4-minute diffusion, see in the spectra in Supplementary Fig. 32. After the diffusion the residual powder on the surface was wiped away.

Supplementary Figure 32. Enhanced absorption during dying. (a) UV-vis spectra after thermal diffusion of Dispersed Red 1 and (b) the change of absorbance at the excitation wavelength (532 nm). (c) UV-vis spectra after thermal diffusion of Dispersed Blue 14 and (d) the change of absorbance at the excitation wavelength (635 nm). Sample thickness: 0.1 mm. Thermal diffusion is processed at 100 °C.

3- Figure 1 is critical for explaining the core concept of the study. However, its current format is cluttered and challenging to follow. To enhance clarity, I recommend splitting it into two separate figures: Figure 1: Focus on the mechanisms (e.g., a-f). Figure 2: Highlight the experimental results and key data (e.g., h to g). This separation will improve readability while maintaining the logical flow of the concept.

Our answer: We thank you for the constructive comments. After careful consideration, we have decided to group the schematic drawing in New Figure 1 and illustrate the oscillation data in New Figure 2, to obtain a better illustration of the concept and comparison between the two oscillation models. Now, they look,

Figure 1. System concept.

Reviewer #6:

The manuscript titled “Light communicative materials” proposes the use of light to couple photothermally deformable objects such as LCE films. An opaque, non-responsive object is attached to the of a photo-responsive object to form a basic unit that is defined as operator. Experimental and numerical results are presented for one self-coupled operator and for two mutually coupled operators. A self-coupled operator exhibits self-oscillation and two mutually coupled oscillations exhibit coupled oscillatory dynamics. Further perturbations (such as wind and light) are applied to one of the mutually coupled operators, and such perturbations lead to the disturbance of the coupled dynamics of the second operator. The coupling is experimentally demonstrated for various photothermally deformable objects and the proposed light-based coupling is further exploited to demonstrate adaptation, control and sensing applications of the coupled operators. In simple terms, this manuscript proposes a method to replace physical springs with a laser in coupling objects.

Our answer: We are grateful for the careful reading and the comments.

Although the proposed concept seems interesting and the demonstrations are impressive, the authors seem to overclaim the significance of the work and the comparison to biological systems seem rather superficial. Additionally, there are many inconsistencies among the experimental data for the same scenario and the proposed analytical model is not fully convincing (see detailed comments below). The manuscript is very cumbersome to read. A thorough and careful rewriting of the manuscript will be helpful in conveying the message concisely. With the above comments in mind, the experiments and the proposed concept of light-based coupling is interesting and offers some advantage over mechanical coupling of objects (such as longer range and smaller delays). Therefore, the paper may be considered for publication after the following comments are addressed:

Our answer: We are grateful for the comments. We also thank the reviewer for commenting, “*the experiments and the proposed concept of light-based coupling is interesting and offers some advantage over mechanical coupling of objects (such as longer range and smaller delays).*” Please see below our point-by-point response to the reviewer’s general concerns.

About the overclaim of the significance. We acknowledge that it was our mistake to overemphasize the results, which may have led to unintended misinterpretation. We have thoroughly reviewed the manuscript to ensure accurate scientific descriptions and to communicate the limitations of the system. Additionally, we have removed the superficial analogy to biological systems.

About the inconsistencies between the experimental data and the unconvincing model. We would like to note that this study focuses on soft matter systems, where soft mechanical actuators are inherently less reproducible, in terms of absolute oscillation frequency, deflection, amplitude, etc., than conventional metallic cantilevers. Achieving stable self-oscillation requires careful tuning of multiple parameters, including excitation spot position, incident angle, and the delay distance between the beam spot and the initial baffle position. These parameters can vary across different optical alignments, influencing the absolute values obtained in different experiments, as well as affecting the accuracy of model fitting. We provide detailed technical responses in a later section. In the revised manuscript, we have also included a comprehensive and self-critical discussion of the system's limitations.

A cumbersome manuscript. The original manuscript was intended to present a method for light-mediated interaction between materials. Following the first round of review, the authors were advised to incorporate additional content, leading to the inclusion of (1) a new figure illustrating adaptation, and (2) a new figure demonstrating a robotic control application. We believe these additions may have contributed to a more cumbersome reading experience. In the revised version, we have made the following changes to improve clarity and focus:

1. Moved the figure related to robotic control to the Supplementary Note 2.4, as robotic control is a well-established concept and not central to this study.
2. Split the original Figure 1 into two separate figures for clearer illustration.
3. Expanded the discussion on adaptation to better highlight how this work differs from previous studies in the literature.

The revised manuscript now follows a logical progression: single self-oscillator (conventional approach) → coupled self-oscillators (novel approach by this study) → oscillator network (adaptation in both shape-morphing and rhythms) → concept generalization.

1) In line 48, what do authors mean by mutual interaction mainly through contact-based methods? There are known physical interactions that are not contact-based, such as hydrodynamic interactions and magnetic interactions.

Our answer: We are grateful for the insightful comments. The term “contactless methods” refers to interactions mediated by fields, such as optical fields. In contrast, hydrodynamic and aerodynamic interactions involve the exchange of substances (liquids or air) or momentum between an object and its surroundings, and are therefore still considered to be physically contact-based methods.

In our view, magnetic interactions should not be considered within the context of this study. Self-oscillators operate based on non-equilibrium, dissipative mechanisms. There is a fundamental distinction between magnetically driven systems, which rely on external forces, and energy-driven oscillator systems. The latter, for instance, a photothermal effect induced by a light beam, delivers energy to the actuator couplers or network without imparting external force or momentum. In other words, a static magnetic field cannot generate self-oscillatory behaviour under non-equilibrium conditions – but, a constant light beam can, as illustrated in this study. While a modulated magnetic field can induce periodic configurational changes, this approach falls under conventional techniques involving external field control and micro object manipulation, as exemplified by magnetically controlled microrobots in the literature.

In contrast, the self-oscillatory behaviour observed in our study arises spontaneously and does not rely on external triggers. We have revised the manuscript accordingly for improved clarity. Now, it reads,

These research efforts are conceptually distinct from field-modification mechanisms, for example, magnetically driven microrobots, whose actuation properties are entirely governed by external field control. Life-like matters operated far from equilibrium underscore the autonomy and adaptive nature of interactive constructs, paving the way for new trends in research grounded in dissipative mechanisms.³⁷

We also attached the cover letter for the original submission at the end of the response letter for a better explanation about the motivation of this study.

2) *The claim in lines 56-57 (A contactless approach...emulating biological systems...) seems superficial and unsupported. Why is low temporal delay required to emulate biological communication at cell-level (the reference cited for this line is on cell biology).*

Our answer: We are grateful for this constructive comment. Our initial intention was to highlight the significance of long-distance communication by drawing an analogy with cellular signalling mechanisms.

REDACTED

However, upon reflection, we recognize that our initial analogy oversimplified the complexity of biological systems and was not scientifically rigorous. We have therefore revised the manuscript accordingly and removed the reference to cellular signalling in this context. Now, it reads,

A contactless approach facilitating low temporal delay, spatial coverage across long distances, and precise directional control offers promising potential for realizing life-like artificial systems with programmable interaction.

3) *The setting (and definition) in Fig. 1d seems opposite to that shown in supplementary fig. 4.*

Our answer: We are grateful for careful reading. The definitions of the negative operator are the same between two figures. However, the initial bending direction in the schematic drawing was placed in the opposite to one in the experiment, which misleads the readers. We have modified the Supplementary Figure 4 to provide a better illustration. Now, it looks,

4) Why is the comparison of model and experiments not shown for a single self-oscillator in fig. 1e?

Our answer: Thanks for the constructive suggestion. We have added the theoretical prediction into the figure. Now, it looks,

Figure 2. From single self-oscillator to coupled oscillators. (a) Schematics illustrating an optical feedback system through mirror reflection. The actuator is activated by the reflected laser beam that is controlled by the operator itself. Top: relaxed state (light on). Bottom: active state (light is cut off). (b) Variation of amplitude and period of single oscillator upon increase of excitation power. Light: 2 mm in diameter, 532 nm continuous laser. **Solid lines represent the simulation results. Details of the modeling and fitting parameters are provided in Supplementary Notes 2.1 and 2.2.**

5) In Fig. 1g, why is the deformation of -ve operator always positive but +ve operator is oscillating symmetrically about 0?

Our answer: We thank you for the careful reading. To synchronously record the movements of two coupled oscillators, we used a single camera to capture both oscillators simultaneously. Positional tracking was then performed concurrently for both. As a result, a displacement offset

always exists between the two oscillators. This offset can vary depending on the optical pathway and the specific experimental setup. We have added a few sentences in the *Methods* section to provide a clearer explanation.

Positional tracking was performed simultaneously for all oscillators. A coordinate offset exists between samples, and its value depends on the experimental conditions.

6) *Inconsistencies in the data:*

a. *The data in supplementary Fig. 2.3 does not match with that shown in Fig. 1e. In the supp. fig, the amplitude saturates around 1.5 mm after 500 mW, but in Fig. 1e, the amplitude keeps increasing above 1.5 mm, even at 1000 mw.*

b. *plot in Fig. 1g is inconsistent with results in supp fig. 4d (bottom) and supp fig. 6c. The displacement curves don't match. Why is the data so different (period is 1 s in fig. 1 but around 1.7 s in supp fig. 4d)? The amplitudes and period are higher than the theoretical prediction and they don't fit the predicted trend. The data in supplementary fig. 6c does not seem sinusoidal.*

c. *The amplitude for -ve operator is around 1 mm but for the same laser intensity in Fig. 1h, the amplitude is around 4-5 mm. Why?*

d. *Colour coding of the data in Fig. 1g can be made consistent with that of the images in Fig. 1f for better clarity.*

e. *In lines 137-138, text refers to green and red LCEs in Fig. 1c, but the Fig. shows blue and red LCEs.*

f. *Why is the experimental data in supp. fig. 8c inconsistent with the data in Fig. 1h-i?*

g. *Why does the data in suppl. fig. 23b not match with the data shown in Fig. 3e even though they are from the same experiment?*

Our answer: We thank you for the careful reading!

(a) The dataset in Supplementary Fig. 2.3 is identical to that in the original Figure 1e. However, the supplementary figure was plotted with a range of 0–500 mW, whereas the main figure covers double that range. We have updated both figures to ensure consistency.

(b) The experimental data plots in the original Fig. 1g, supplementary figs. 4d and 6c are using very different laser powers. They are, original Fig. 1g: Laser 1, 532 nm, 138 mW; laser 2, 635 nm, 210 mW. Supplementary Fig. 4d: laser 1, 532 nm, 280 mW; laser 2, 635 nm, 320 mW. Supplementary Fig. 6c: laser 1, 532 nm, 390 mW; Laser 2, 635 nm, 460 mW. This is why they exhibit different amplitudes and frequencies.

The theoretical prediction trend shows the change of amplitude and frequency upon an increase in only one laser power, when the other laser power is fixed. The theoretical prediction and experimental data shown in the original Fig. 1h (New Figure 2e) are based on laser 1 power = 180 mW, while the experiments in original Fig. 1g and supplementary fig. 4d are not. This is the reason, the amplitude and frequency shown in these two figures don't fit into the theoretical prediction curve.

The oscillator in this study belongs to the category of relaxation oscillator. van der Pol equation (https://en.wikipedia.org/wiki/Van_der_Pol_oscillator) is generally used to describe such an oscillator. Usually, the oscillation curve does not show a sinusoidal shape, unless strong nonlinearity exists. As exhibited by most of the figures in this manuscript, non-sinusoidal

oscillation curves were usually observed. To avoid misleading, we have changed the original Figure 1g, new Figure 2d into a non-sinusoidal one. Now it looks,

We would like to note that, the oscillating curve in New Figure 2 (d) is corresponding to the data point (laser 1, 180 mW; laser 2, 150 mW) in Figure 2(e).

In the revised version, we also explored the influence of delay distance on the change of waveform. A discussion is added into the Supplementary Note 2.3, now it reads,

However, with the mechanical properties held constant, variations in delay distance w_0 – the minimum deflection distance of the baffle required to trigger the cut-on/-off action – can significantly influence the oscillation behavior. Figure 2.8a presents simulated results showing that increasing w_0 at one of the operators reduces the oscillation frequency and affects the waveform.

Figure 2.8 Influence of delay distance on the oscillation property. (a) Waveform and (b) phase delay of two coupled oscillators by change of the delay distance in the negative operator (w_{0-}). In the computation, we set $l_- = 2.5$ cm, $l_+ = 2.5$ cm, $w_{0+} = 4.37$ mm, $\beta = 0.8$, $\tau_{inertial} = 0.015$ s, $\tau_{heat} = 0.3$ s, $P_- = 106$ mW, $P_+ = 360$ mW, $\lambda_- = 0.0017$ /mW, $\lambda_+ = 0.001$ /mW.

(c) In the original Figure 1g, the laser 1 power is 138 mW, and laser 2 is 210 mW. In the original Figure 1h, all the data set is recorded by fixing the laser 1 power at 180 mW, distinct from the setting of Figure 1g. This is the reason the oscillation data exhibit different frequencies and amplitudes. We have changed the figure accordingly, as explained in the previous answer.

(d) We appreciate the suggestion. The figure has been modified accordingly.

(e) Our apology for the typo. We have changed it accordingly.

(f) The amplitude data in Supplementary Fig. 8c is identical to that in the original Fig. 1h (now Fig. 2e). However, the period data in Supplementary Fig. 8c is inconsistent with the main figure. We sincerely apologize for this significant error. During manuscript preparation, we plotted data for both the oscillation period and the time delay between the two oscillators, as shown in the figure below. However, the phase delay data was highly sensitive to environmental conditions and varied significantly depending on the experimental setup. As a result, we were unable to draw a definitive conclusion regarding the relationship between phase delay and other experimental conditions. To simplify the narrative, we decided to remove the time delay data. Unfortunately, the periodicity data was mistakenly deleted instead. We have corrected the figure accordingly.

Raw data of variation of the period and delay time with excitation power.

New Supplementary Figure 8. Light power dependent behaviors in the coupled oscillators.

(g) In our opinion, two data, now presented in New Figure 5e and Supplementary Fig. (new Supplementary Fig. 27), match each other very well when the disturbance is not applied. Both data show the same amplitude, frequency and waveform. The disturbance used in the main figure is LED illumination, during which the oscillation does **not** cease. However, the disturbance used in the supplementary figure is a mechanical **cessation**, thus bringing different influence on the oscillation. To avoid misunderstanding, we have modified the text, now it reads,

Additional data on disturbed oscillation resulting from **external mechanical cessation** can be found in Supplementary Fig. 23.

7) Does the phase difference/time delay between two coupled operators depend on the size of the baffle and thus the minimum deformation of the LCE required to block/unblock the laser spot?

Our answer: Thanks for the constructive comment. Yes, the phase delay depends on the minimum deflection distance of the baffle. We have added the discussion in the Supplementary Note 2.3 for a better illustration. Now, it reads,

Figure 2.8 Influence of delay distance on the oscillation property. (a) Waveform and (b) phase delay of two coupled oscillators by change of the delay distance in the negative operator (w_{0-}). In the computation, we set $l_- = 2.5$ cm, $l_+ = 2.5$ cm, $w_{0+} = 4.37$ mm, $\tilde{\beta} = 0.8$, $\tau_{\text{inertial}} = 0.015$ s, $\tau_{\text{heat}} = 0.3$ s, $P_- = 106$ mW, $P_+ = 360$ mW, $\lambda_- = 0.0017$ /mW, $\lambda_+ = 0.001$ /mW.

The theoretical analysis also indicates that the change of w_0 also influences the phase difference between two coupled oscillators. Figure 2.8b presents simulated results showing that an increase of w_0 reduces the phase delay from near 180 to 20° at high w_0 . However, in the experiments, the phase delay highly depends on the irradiation power and sensitive to environmental fluctuations. Comparison between experimental and simulated results is challenging in this study.

8) In suppl. fig.3, the curvature is defined as $1/\text{radius}$. The radius cannot be negative, then how is curvature negative?

Our answer: Thanks for the comment. The sign of the curvature indicates the direction of the bending. The LCE strip usually has an initial bending after photopolymerization. Upon heat elevation, it bends into a flat shape (curvature = 0), then bends toward the opposite side. A negative curvature is also used in mathematics.

9) Why is the data in suppl. fig.3 presented for multiple trials on only 1 sample? How much does the statistics vary over different samples? and would such variation affect the main results of the paper?

Our answer: Thanks for the constructive comment. We have performed additional experiments to measure the bending performance of three LCE strip samples cut from the same liquid crystalline film. The result is shown in the new Supplementary Figure S3.

Supplementary Figure 3. Stimuli-responsiveness of LCE actuator. (a) Left: Photographs showing the LCE strip geometries at room condition and elevated temperature. Strip size: $15 \times 1 \times 0.1 \text{ mm}^3$. The sample is placed on top of a hot plate and covered with a transparent glass window to attain homogeneous temperature distribution. Right: The curvature variation upon increasing the temperature. Curvature is defined as $1/r$, where r is the radius of the strip, as shown in the inset. (b) Left: Photographs displaying the LCE strip geometries upon different illuminating intensities (635 nm, laser source). Strip size: $15 \times 1 \times 0.1 \text{ mm}^3$. Right: The bending angle of three independent LCE actuators upon change of illuminating light intensity I . The strips are cut from the same LCE film. The bending angle (α) is indicated in the inset. Error bars represent s.d. for $n = 3$ measurements. The same sample was measured repeatedly. All scale bars are 1 cm.

We would like to note that this study focuses on soft matter systems, where soft mechanical actuators are inherently less reproducible than metallic cantilevers. As shown in the figure above, even strips of identical size, cut from the same LCE film – which was photopolymerized in one single UV exposure – can exhibit variations in deformability. This variability arises from the spontaneous self-assembly nature of liquid crystals, which inherently introduces randomness into the samples. Consequently, reproducing self-oscillations with identical frequency, amplitude, and waveform is particularly challenging in soft matter actuators.

We would also like to emphasize that the **main results** of this manuscript are:

- (1) the development of a model for inducing both positive and negative feedback through photomechanical actuation;
- (2) the demonstration that coupled oscillators can self-oscillate under constant laser irradiation,

thereby operating far from equilibrium;

(3) the observation of signal transmission between coupled oscillators; and

(4) the ability of the system to change both shape and frequency in response to external disturbances, enabled by the feedback network.

Therefore, variation between samples will not affect the main findings of this manuscript; a detailed characterization of the mechanical properties of the soft LCEs, as well as an in-depth analysis of the oscillation kinetics, lies **outside the scope** of this study.

10) In line 117, what is the threshold for laser power needed? what factors does this threshold depend on?

Our answer: The power threshold refers to the minimum laser power required for the actuator to deform sufficiently to interrupt the laser beam, thereby introducing negative feedback. If the laser spot is positioned farther from the edge of the baffle, greater deformation is needed to reach the beam path, which in turn requires higher laser power to initiate self-oscillation. We have revised the manuscript to clarify this explanation.

A power threshold is required to deform the LCE and move the baffle closer to the laser spot, thereby initiating the negative feedback mechanism. Beyond such threshold, the operator undergoes self-oscillation fueled by a continuous light beam

11) In suppl. fig. 10, why does the - operator not revert back to its original oscillation curve? There is a positive offset in the curve after the disturbance is removed.

Our answer: Thanks for the comment. LCEs are soft matter, often exhibit large strain in deformation, and thus inherently less reproducible than metallic cantilevers. The viscoelastic nature of soft polymer results in a drift of oscillation date, after the sample being illuminated for a long time and experiencing multiple deformation cycles. In the revised manuscript, we have provided a discussion about this. Now they read,

Unlike thermo-responsive cantilevers made of metallic bilayers, soft actuators composed of LCEs exhibit non-uniform deformation along the strip during dynamic oscillation. Supplementary Fig. 31 presents a series of photographs capturing the shape-morphing behaviour of the soft LCE in the experiments, revealing a wide variety of configurations. These variations arise from differences in sample orientation and the position of the laser excitation spot relative to the strip in different experimental setups. Additionally, the light-heat-induced deformability varies among samples due to the self-assembled nature of liquid crystals – even strips cut from the same film prepared in the same cell can exhibit subtle differences in actuation behaviour (Supplementary Fig. 3b). The viscoelastic properties of the soft material further contribute to a gradual drift in oscillation behavior over extended periods, as observed in the Supplementary Figs. 4, 10, 14, 16, 17, 29.

12) How robust are the oscillator dynamics to external perturbations? Does the dynamics/coupling change after multiple perturbations, or after a strong disturbance?

Our answer: Thank you for the valuable comments. Supplementary Figs. 10, 11, 14-17, 22, 27 present the oscillation data recorded before, during, and after the perturbations were applied

to the coupled oscillators. Across all datasets, the oscillators consistently return to their initial states, exhibiting similar frequency, amplitude, and waveform after the perturbations are removed.

In some scenarios, strong disturbances were introduced to temporarily cease the oscillations (Supplementary Figs. 16, 17, 22, 27). However, these did not affect the eventual revival of the system once the disturbance was lifted. Additionally, multiple types of perturbations, including wind (Supplementary Fig. 16), LED light (Supplementary Fig. 14), and heat (Supplementary Fig. 15), of varying strengths were applied to the same coupled oscillators within a single experiment. These were recorded as multiple perturbation phases, further emphasizing the robustness of the system.

The observed drift in the oscillation data is a known characteristic linked to the inherent properties of soft matter systems, as previously explained in our earlier response.

13) The coupling strength/communicated information is not quantified for any case. Such quantifications may be helpful in better understanding the strength of the work.

Our answer: Thanks for the comment. We would like to note that the quantified analysis of oscillation data between two oscillators and the transmission of information between two soft matter strips are not the main focus of this study. To avoid misunderstanding, we have examined and rephrased the text avoiding using “communication”. We have also changed the manuscript title to be,

Light-mediated interactions in responsive materials: From individual self-oscillators to feedback-driven network.

About the coupling strength, we have explained the difference between the reported oscillators with existing ones in the literature. We have modified the text for a better illustration.

Depending on the coupling strength, a disturbance to one oscillator can have varying effects on the other.⁵³ In this study, we present a method to achieve strong coupling between two oscillators using light beams – such that if one oscillator ceases, the other also stops. This behavior is distinct from that observed in existing self-oscillating systems (see Supplementary Fig. 13). This method also offers designs of positive and negative operators that can be programmed to establish diverse feedback loops among multiple oscillators.

14) There are several concerns regarding the model:

Our answer: We are grateful for the reviewer’s careful reading and valuable comments.

a. The only direct dependence on the laser power in the model appears for $A=a(1-\exp-P/P_0)$. Given this assumption, it is not surprising at all why the deformations follow the same curve suppl. Fig. 2.3. Can you justify why this assumption is made? What is the physical motivation for such assumption? Does the data in Fig. 3b support this assumption?

Our answer: Thanks for the comment.

Justification. When we extended the illumination intensity range in the photo-induced bending experiment (as shown in the original Supplementary Fig. 3b), saturation behavior was observed, as illustrated in the figure below. This saturation is attributed to a self-shadowing effect—when the strip bends significantly, the front portion begins to block part of the incident light.

Figure. The bending angle of the LCE actuator upon change of illuminating light intensity, I . Error bars represent s.d. for $n = 3$ measurements. The same sample was measured repeatedly. Left: original supplementary figure 3 in the previous submission. Right: light induced bending curve with larger range of illumination intensity.

This has been the reason we assumed a nonlinear relationship between light induced deflection and input laser power, and used a form of $\bar{A} = a(1 - e^{-P/P^0})$.

Influence in the modelling. We would like to clarify that the saturation curve shown in Supplementary Fig. 2.3 is **not** the result of exponential fitting. This type of saturation behavior is a typical characteristic of self-oscillatory systems. Additionally, we note that our original submission employed only a **simplified linear model**.

In illuminated state, i.e. $\bar{w} \leq \bar{w}_0$:

$$\bar{w}_L(\bar{t}) = \bar{I}(1 - e^{-\bar{t}/\bar{\tau}_{\text{heat}}}), \quad (14)$$

in non-illuminated state, i.e. $\bar{w} > \bar{w}_0$:

$$\bar{w}_L(\bar{t}) = \bar{I}e^{-\bar{t}/\bar{\tau}_{\text{heat}}}, \quad (15)$$

where the dimensionless light intensity is defined as $\bar{I} = \lambda I$, with $\lambda = \frac{A\eta}{kt}$ denoting the deflection coefficient of light-driven bending.

This simple model yields a prediction that qualitatively agrees with both the experimental and simulated data.

In the first round of review, Reviewer 2 recommended that we provide a quantitative model. That has been the reason, we used a nonlinear model in the second-round submission. However, as noted in our earlier replies (#9 and #11), developing a comprehensive quantitative model remains a significant challenge in LCE soft actuator and is beyond the scope of this study.

After careful consideration, we have decided to revert to a **simplified linear model**, which more effectively captures and clarifies the fundamental physical mechanisms of the system. This revised model is now presented in the updated Supplementary Note 2. The comparison between experimental data and simulated results based on the linear model is shown below,

Figure 2.3. Dependences of amplitude and period of single oscillator on the laser power. In the simulation, we set $l = 2.5$ cm, $w_0 = 5$ mm, $\bar{\beta} = 0.8$, $\tau_{\text{inertial}} = 0.015$ s, $\tau_{\text{heat}} = 0.3$ s, and $\lambda = 0.0026/\text{mW}$. Laser spot size: 2 mm.

b. In the model, the baffle size is not considered (baffle is implicitly treated as a point object). Therefore, why does the operator require a critical deflection in the simulations to result in oscillation? Why is such critical deflection needed in the experiments?

Our answer: We acknowledge that it was an oversight on our part to overlook this issue. The displacement of the baffle and the deflection range required to induce the cut-on/cut-off action are determined by the boundary conditions, and are independent of the schematic shape shown in the figure. We have provided details in the Supplementary Note:

For single oscillator:

in illuminated state, i.e. $\bar{w} \leq \bar{w}_0$:

$$\bar{w}_L(\bar{t}) = \bar{P}(1 - e^{-\bar{t}/\bar{\tau}_{\text{heat}}}), \quad (14)$$

in non-illuminated state, i.e. $\bar{w} > \bar{w}_0$:

$$\bar{w}_L(\bar{t}) = \bar{P}e^{-\bar{t}/\bar{\tau}_{\text{heat}}}, \quad (15)$$

where $\bar{P} = \lambda P$, with $\lambda = \frac{A\eta}{kl}$ denoting the deflection coefficient of light-driven bending.

For coupled oscillators,

$$\bar{w}_{L-}(\bar{t}) = \begin{cases} \bar{P}_- \left(1 - e^{-\frac{\bar{t}}{\bar{\tau}_{\text{heat}}}}\right), & \bar{w}_+ > \bar{w}_{0+} \\ \bar{P}_- e^{-\frac{\bar{t}}{\bar{\tau}_{\text{heat}}}}, & \bar{w}_+ < \bar{w}_{0+} \end{cases}, \quad (19)$$

$$\bar{w}_{L+}(\bar{t}) = \begin{cases} \bar{P}_+ \left(1 - e^{-\frac{\bar{t}}{\bar{\tau}_{\text{heat}}}}\right), & \bar{w}_- < \bar{w}_{0-} \\ \bar{P}_+ e^{-\frac{\bar{t}}{\bar{\tau}_{\text{heat}}}}, & \bar{w}_- > \bar{w}_{0-} \end{cases}, \quad (20)$$

in which, \bar{w}_{0-} and \bar{w}_{0+} denotes the on/off transition critical deflections of LCE cantilevers (-) and (+), respectively.

We have also updated the schematic diagram to improve clarity.

c. Why is the force from laser power modelled as a spring force? What is the physical motivation for the choice? Does a single beam oscillate on constant illumination? It does not appear to be the case.

Our answer: As shown in Figure 2.1 c and d, the total deflection involves the elastic deflection and the light-driven deflection. For the total deflection at any time, the light-driven deflection affects the elastic deflection, and in turn determines the bending force. To make it clearer, we have replotted Figure 2.1 and renamed the force from laser power to bending force in the revision.

Figure 2.1. Schematics of a feedback self-oscillator. The oscillator is composed of an LCE cantilever and a baffle.

Dynamics of the single oscillator. Figure 2.1 illustrates the single oscillator, consisting of an LCE beam, a baffle, a laser beam, and a mirror. Initially, the baffle allows the propagation of light, and the laser beam induces bending in the LCE through photothermal actuation (Figure 2.1 a). Consequently, due to LCE deformation, the baffle impedes the light beam (Figure 2.1 b). Subsequently, the light-induced bending rebounds, causing the LCE cantilever to unbend, thereby allowing the resumption of light propagation and initiating a new cycle. During the vibration, the baffle is subjected to the bending force F_b of the LCE beam, and the damping force F_d (Figure 2.1 d), therefore the governing equation for its vibration is written as

$$m \frac{d^2 w(t)}{dt^2} = F_d + F_b, \quad (1)$$

where m is the mass of the baffle, $w(t)$ is the end deflection of the LCE cantilever. For simplicity, the damping force is assumed to be proportional to the velocity of the baffle, *i.e.*,

$$F_d = -\beta \frac{dw(t)}{dt}, \quad (2)$$

in which, β is damping coefficient. For simplicity, the bending force of the LCE beam is assumed to be proportional to the elastic bending deformation, *i.e.*

$$F_b = -\frac{3\pi}{l^3} w_e(t), \quad (3)$$

in which, π is bending stiffness, l is the length of the LCE cantilever. As shown in Figure 2.1 c, the elastic bending deformation $w_e(t)$ depends on both the light-driven bending deflection $w_L(t)$ and current bending deflection $w(t)$ (*i.e.* total bending deflection), which is calculated as

$$w_e(t) = w(t) - w_L(t). \quad (4)$$

We would like to emphasize that the concept presented in this study is based on self-oscillation, where a single beam self-oscillates when illuminated by a **constant** laser, or multiple structures self-oscillate when excited by multiple constant laser beams. To initiate oscillation, a negative operator must be introduced into the loop to induce negative feedback. The oscillation is self-regulated by the responsive material itself and does **not** require temporal modulation of the external excitation beam.

d. How realistic are the chosen simulation parameters? For example, is the choice of $P_0 = 2-3$ W physically meaningful? Please report the values of all the parameters of the model that were used for all the calculations.

Our answer: In the nonlinear model, P_0 is used to numerically fit the saturation behavior based on the exponential assumption. Therefore, the value of P_0 is not directly associated with a

specific physical variable. As explained in response #14(a), we have reverted to the linear model to better capture and clarify the underlying physical mechanisms. All fitting parameters used in the model are reported in the revised version. For better clarification, we have added two tables to list all the parameters for the modelling.

Table 1 Material properties and geometric parameters.

Parameter	Definition	Value	Unit
l	Length of the LCE cantilever	2.5	cm
w_0	Critical deflection (delay distance)	5	mm
$\bar{\beta}$	Dimensionless damping coefficient	0.8	/
τ_{inertial}	Inertial characteristic time	0.015	s
τ_{heat}	Heat time scale	0.3	s
λ	Deflection coefficient of light-driven bending	0.0026	/mW
P_0	Laser power	0~1000	mW

Table 2 Material properties and geometric parameters for coupled oscillators.

Parameter	Definition	Value	Unit
l_-	Length of the LCE cantilever -	2.5	cm
l_+	Length of the LCE cantilever +	2.5	cm
w_{0-}	Critical deflection of the cantilever (delay distance) -	0~5	mm
w_{0+}	Critical deflection of the cantilever (delay distance) +	0~5	mm
$\bar{\beta}$	Dimensionless damping coefficient	0.8	/
τ_{inertial}	Inertial characteristic time	0.015	s
τ_{heat}	Heat time scale	0.3	s
λ_-	Deflection coefficient of the LCE cantilever -	0.0017	/mW
λ_+	Deflection coefficient of the LCE cantilever +	0.001	/mW
P_-	Laser power of laser beam (-)	0~1000	mW
P_+	Laser power of laser beam (+)	0~1000	mW

e. It is not clear what causes the slow dynamics of the coupled oscillators compared to the single oscillator. It would be helpful to add a simple physically meaningful explanation. Lines 147-148 are not clear and seem superficial. In the model this delay is somehow hardcoded by the conditions on displacements.

Our answer: Thank you for this valuable comment. In all self-oscillating systems, there is an inherent delay between the material's stimulus and its response. This delay results in a residual force at the equilibrium position, which drives the material to continue deforming and move away from equilibrium. In the case of coupled oscillators, the two LCEs undergo alternating oscillations. The delay is governed by the interval between one actuator turning the light on and the other turning it off, or vice versa. We have added a concise explanation in the main text and revised Supplementary Fig. 7 by including a schematic diagram aligned with the oscillation curve to better illustrate this concept. Now, they look,

The delay, denoted as t_d in Supplementary Fig. 7, represents the time required for the two baffles to transit alternately at their on and off positions. This delay is also reflected in the time gap between the state changes (on/off) of the two laser beams. The comparison between the single oscillator and coupled oscillator and the equivalence model for kinetics illustration, see in Supplementary Note 2.3.

Supplementary Figure 7. Displacement kinetics explanation. In the experiment, the parameters are Laser 1: 180 mW, laser 2: 150 mW, $l_- = 2.5$ cm, $l_+ = 2.5$ cm. t_d , is the time delay. t_{d1} : time between laser 1 ON and laser 2 OFF. t_{d2} : time between laser 2 ON and laser 1 ON. t_{d3} : time between laser 1 OFF and laser 2 ON. t_{d4} : time between laser 2 OFF and laser 1 OFF.

For a better understanding of such delay, we have also added an equivalence model for comparison between single and coupled self-oscillators. Now, it reads,

Equivalence model. A light-fueled self-oscillator is a mechanical structure that oscillates under a constant light field. In both single-oscillator and coupled-oscillator systems, a negative feedback loop governs the self-oscillation process. Figure 2.9a illustrates this mechanism through an equivalent model, highlighting the kinetics of the negative operator.

In the single self-oscillator, the mechanism is relatively straightforward. The beam deflects from its equilibrium position to a critical distance w_0 , at which point it blocks the incident light that excites itself. This interruption acts as a negative feedback signal, temporarily turning off the excitation and later on turning on it, sustaining continuous oscillation.

In the coupled oscillator system, the beam also deflects to a critical distance and blocks the incident light at w_0 . However, this blocking does not directly turn off the light exciting the same operator. Instead, it first extinguishes the light exciting the positive operator. Once the positive operator is turned off, it in turn blocks the other light beam, thereby switching off the light exciting the negative operator. This requires the baffles on both operators always travel for a longer distance than w_0 , introducing an additional time delay dependent on the deflection distance and the photothermal response time of the material. A similar delayed mechanism occurs when the baffle returns to switch the light back on.

Figure 2.9 Equivalence model for coupled oscillators. Kinetics of (a) single self-oscillator and (b) coupled oscillators with time delay.

15) *The conversion of deformation to electric signal is creative and could be useful. However, In Fig. 5a, why is the shape of the measured signal different from the shape of the deformation, especially because the deformation is linearly related to the incident power (also assumed in the model)?*

Our answer: In the coupled oscillator system, each operator is equipped with a baffle that performs either a cut-on or cut-off action. As a result, the light intensity transmitted from one oscillator and measured near the other appears as an on–off signal. This means that the measured light power is not linearly dependent on either the incident power or the deformation. The nonlinear response of the photodiode, combined with the scattered light from the aluminum baffle near the photodiode, contributes to the dome-shaped waveform observed. We have added explanation in the revised Supplementary Note 2.4. Now, they read,

The opening and closing actions of the baffle in operator 1 transmit an on-off signal of light intensity.

Note that both the nonlinear response of the photodiode and the scattered light from the nearby aluminium baffle affect the waveform, transforming it from an ideal rectangular shape to one with a domed top.

16) *The definition of adaptation seems very different from the commonly perceived meaning of adaptation and this definition is not backed up with any references. Therefore, the definition appears to be the opinion of the authors and not an established fact. The results on “adaptation” do not seem to add much value to the paper. Many known materials, for example, shape memory alloy, clay, bistable beams, all seem to fit this definition of adaptation. The results of Fig. 4 could be taken out and reported separately after thorough and careful analysis. This reduction will also help to make the paper more concise and easier to read.*

Our answer: We are grateful for these meaningful comments. About the adaptation: In our opinion, the term “adaptation” has been widely misused in the literature. In most case studies, it is used to illustrate the concept of stimulus-responsiveness rather than a true adaptation from the bioinspiration perspective. Our viewpoint on adaptation in responsive materials is that a material possesses a state (such as shape or frequency), which can change in response to external stimuli and maintain the altered state even after the stimuli cease. We have added a discussion to explain this perspective by comparing it to previous literature. It reads,

In previous literature, adaptation has often been described as the responsive shape change of a material in reaction to environmental stimuli. In this study, we propose a broader perspective: adaptation can be viewed as the capacity of a material to change its state in response to external disturbances, coupled with the ability to retain this altered state even after the disturbance is removed.

Respectfully, we disagree with the reviewer’s comment that, “*The results on “adaptation” do not seem to add much value to the paper*”. This manuscript introduces a method to program

both positive and negative feedback in light-responsive materials, enabling the construction of feedback loops. One of the novel functionalities arising from these feedback loops is the demonstration of programmed adaptation behavior – the actuator can change its shape or oscillation frequency in response to varying external perturbations. As such, we believe this figure highlights a core novelty of the manuscript.

We agree with the reviewer's observation that "*Many known materials, for example, shape memory alloy, clay, bistable beams, all seem to fit this definition of adaptation.*" However, we would like to emphasize that preserving deformed geometry in a material is **not a trivial task**. It often requires either specialized materials such as shape memory alloys, reorganization of molecular packing (as in clay), or mechanical structures with built-in energy barriers (as seen in bistable beams) – mentioned by the reviewer – and others, e.g., plastic deformation, viscoelasticity, magnetic/electric response, etc. In contrast, in this soft actuator system, it exhibits bistable shape morphing **without** relying on shape-memory effects, molecular rearrangement, or engineered mechanical bistability, or other effects. This behavior is achieved **purely** through a light-regulated positive feedback mechanism, which we believe represents an unconventional advancement. We have now added a more detailed discussion in the revised manuscript to clarify and support this point.

Typically, maintaining an altered state requires mechanisms, e.g., shape memory effects,⁵⁹ plastic deformation (as in deformable clays), or bi-stable mechanical architectures.⁶⁰ Here, in Figure 4b–g, we demonstrate that a positive feedback loop can induce bi-stability without relying on shape memory, microstructural changes, or specific mechanical designs. Moreover, a material state can be characterized not only by its spatial configuration (the shape) but also by its oscillation frequency in the time domain.³⁷ Thus, adaptation may also refer to the ability to shift and sustain different oscillation frequencies following external disturbances, as illustrated in Figure 4h–k.

17) The figures appear pixelated and images are unclear. Improving the resolution of the images would help in better appreciating the figures.

Our answer: Thank you for the comment. We have uploaded high-resolution figures in the revision submission.

Finally, we are sincerely grateful to the reviewer for the time and valuable feedback, which has significantly helped us improve the quality of our work.

Re: Light-mediated interactions in responsive materials: From individual self-oscillators to feedback-driven network (manuscript NCOMMS-24-40904B)**Point-by-point responses to the Reviewers**

Reviewer #1:

I have followed this manuscript from its first-round submission to Nature Communications and also reviewed critiques and comments from non-listed reviewers in prior rounds of review. Based on the overall comments from the reviewers and the corresponding response from the authors after two-round revisions, I would like to point out that the authors have accepted the criticism regarding material's limited novelty, conclusion overclaim, confusing logic flow, among others, and they revised the manuscript accordingly. This can be seen from the title "Communicative materials" change to a more precise specific one, the adding of a dedicated "system limitations" section to differentiate this work from prior literature, and the extension of the "discussion and conclusions" section. These revisions reflected the authors' positive attitude to improve the quality of the paper, which is good and obviously should be appreciated. However, the acceptance of these criticism raises another fundamental question: Does the revised work meet Nature Communications' threshold for novelty? As a chemist/materials scientist, I recognize that while no new materials were designed, the feedback-driven network oscillation mechanism presents a compelling interdisciplinary contribution, which holds interest for materials science, engineering, and bionics communities. Therefore, I recommend a publication opportunity pending consensus among other reviewers.

Our answer: We are sincerely grateful for the Reviewer's positive and valuable comments. We fully acknowledge the observation that "no new materials were designed." Since both PIs in this study are physicists, our work has not focused on developing new LCE materials in previous publications either. Instead, we see this as a unique perspective in our scientific approach – one that highlights the value of a physics-based methodology independent of a specific type of chemistry or material property. We also wish to express that it has been a truly rewarding experience to engage with the Reviewers through several rounds of review. In our view, this exchange has fostered mutual understanding between chemists, physicists and researchers across broader disciplines.

Reviewer #5:

The authors have adequately addressed all of my questions and concerns. I therefore recommend this manuscript for publication in Nature Communications.

Our answer: We are grateful for the positive comment. We thank you for the great effort in the review process.

Reviewer #6:

Reviewer #6:

I appreciate the revisions made to the manuscript in response to my previous comments. The quality and clarity of the manuscript seems to be improved. However, some concerns regarding “contact-based interactions”, “adaptation”, robustness to external perturbations, and generality and scalability of this work remain and it is important that these concerns are addressed via concise discussions and rewriting to clarify the real significance of this work to a broader research community and to avoid largely underestimating the existing literature.

Our answer: We are grateful for the Reviewer’s scientific rigor and strong sense of responsibility. We truly appreciate the time and effort devoted to providing such thoughtful and detailed comments. We sincerely thank the Reviewer for these valuable suggestions. Below, we provide our point-by-point responses.

1. Contact-based interactions: The authors consider hydrodynamic interactions as contact-based, however these interactions do not require a direct physical contact between two interacting objects, only the contact between an object and its surrounding fluid is necessary. The way authors define “contactless methods” seems to eliminate every possible interaction mechanism and only leave interactions mediated through electric, magnetic fields and electromagnetic waves. Even in those, the authors eliminate the field-mediated interactions.

Our answer: The intention behind categorizing hydrodynamic forces as contact-based interactions was to highlight the fact that momentum is transferred from one object, through the medium (liquid or gas), to another object – essentially a physical interaction mediated by the medium. This process is analogous to direct mechanical contact between two objects without an intervening medium. Ultimately, the distinction between contact and non-contact is largely terminological, and we did not intend to engage in such a debate. To avoid potential misunderstanding, we have removed all references to contact or non-contact definitions in the revised text, focusing solely on the underlying scientific concepts. Now they read,

In the realm of bioinspired material research, the utilization of out-of-equilibrium²⁰ soft matter has enabled versatile life-like functions^{21,22} and novel opportunities for mutual interaction between entities.^{14,23-26}

An ~~contactless~~ approach facilitating low temporal delay, spatial coverage across long distances, and precise directional control offers promising potential for realizing life-like artificial systems with programmable interaction.

~~For example, the work presented includes methods for mechanical interaction in synchronization,²³ self-oscillation driven by negative feedback from heat diffusion,^{12,13} and spatial patterns induced by feedback from chemical reactions.³⁹ In all these systems, feedback plays a crucial role in maintaining the system far from thermodynamic equilibrium, enabling interaction. However, the predominant methodologies rely on contact-based approaches, such as mechanical interaction²³ and the exchange of reactive substances.⁵⁴ These approaches are limited by temporal delays, short transfer distances, and a lack of directionality.~~ The method reported in this study couples two or more self-oscillators through an optical feedback loop, providing high-directionality in network connections and enabling material interaction over long distances with minimal delay.

Respectfully, we would like to clarify that we did not intend to dismiss other interaction mechanisms or underestimate the contributions of previous work. The primary aim of the manuscript is to introduce the concept of light communication – specifically, light beam interaction between two material pieces, both operating far from equilibrium. Importantly, this interaction is not simply a matter of using a light beam to influence one object over the other.

During the second revision, we intentionally moderated the tone and replaced all instances of “communication” with “interaction,” which we believe may have contributed to the misunderstanding. We have carefully reviewed the text throughout and revised it to avoid any potential misinterpretation regarding the scope or intent of our work.

Title: Light-mediated **communication** in responsive materials **ranging** from individual self-oscillators to feedback-driven network

Abstract. In artificial systems, there are means of interaction between individuals, for instance, mechanical contact, hydrodynamic coupling, and thermal gradient, chemical gradient diffusion and electric/magnetic fields. However, they generally lack high directionality or sufficient interaction ranges.

The argument regarding the exclusion of magnetic interactions is weak. These interactions do not always need the existence of external fields, for example, magnetic dipole-dipole interactions can exist in the absence of external fields. Besides, a “constant light beam” is not static itself, light is oscillating electric and magnetic fields. Finally, a constant electric field can actually induce oscillatory behaviour of a solid object within a fluid (see the literature on Quincke rollers).

Our answer: We fully agree with the Reviewer that both magnetic and electric fields can contribute to material interactions; a constant electric field can indeed induce non-equilibrium and self-propelling behaviors. However, the focus of our manuscript is to introduce a light-interactive method that is: (i) established at non-equilibrium state (not simply force interaction), (ii) highly directional (one unit interacting with another via a single light beam), (iii) effective over long distances (on the scale of laser propagation), and (iv) characterized by minimal time delay (governed by the material’s photothermal response rather than gradient diffusion in the medium). We have carefully revised the text to avoid any unintended impression of excluding the valuable contributions of other research approaches.

From a technical standpoint, we would also like to clarify that the comment regarding, “a “constant light beam” is not static itself, light is oscillating electric and magnetic fields” is not directly relevant to the concept of a self-oscillator. In our context, a self-oscillator absorbs light purely as an energy source, with the key factor being the feedback mechanism. Thus, the phenomenon is independent of electromagnetic field oscillations – light of different wavelengths can induce self-oscillations of the same frequency.

The newly added statement regarding magnetic field driven microrobots is incorrect because the magnetically driven microrobots (including those in ref 30) utilize hydrodynamic interactions and magnetic dipole-dipole interactions between the microrobots and only magnetic driving is not enough. Robots in another reference on microrobot swarm cited by the authors (Ref. 29) actually use light-based (IR light) methods for local interactions.

Our answer: We believe that our originally added sentence, “*These research efforts are conceptually distinct from field-modification mechanisms, for example, magnetically driven microrobots, whose actuation properties are entirely governed by external field control*” is scientifically correct. The actuation in such systems relies on magnetic field control, which directly implements magnetic gradient forces or torques. The resulting complex patterns are further influenced by dipole interactions and hydrodynamics; therefore, the observed “swarm” behaviors, as we fully agree with the Reviewer, cannot be attributed solely to magnetic driving. Nevertheless, to avoid any potential misunderstanding, we have removed this sentence.

Ref. 29 reports on a system of a thousand robotic swarms relying on electronics. The ultimate goal of adaptive materials research is to realize materials that exhibit comparable levels of embodied intelligence based on stimuli-responsive feedback, without the use of any electronics. This is the reason Ref.29 has been mentioned in the text.

If the authors intend to limit the discussion to light (and the work does seem focused on self-oscillators), then I suggest that they simply motivate the need for light-based interactions and its significance and leave out the discussion on so called “contact-based” interactions. The significance of light-based interactions and its advantages over other interaction mechanisms (if any) needs to be clarified (perhaps related to its low temporal delays and long-range) as it is not entirely clear in the current state of the manuscript.

Our answer: We are grateful for the Reviewer’s kind suggestion. In the revised manuscript, we have omitted the discussion regarding contact- versus non-contact-based methods. Instead, we have highlighted the key merits of light-based interactions, namely their ability to generate non-equilibrium states, provide high directionality, operate effectively over long distances, and exhibit minimal time delay.

In my opinion, the current text wrongly undermines the literature on other interaction mechanisms. Therefore, the motivation of this work really needs to be strengthened to convey the significance of using light-based interactions for a broader research community (and not just the community on light-responsive self-oscillators), without wrongly undermining the existing literature on other interactions mechanisms.

Our answer: We have carefully reviewed every aspect of the manuscript, with particular attention to tone, to ensure that it conveys a purely scientific discussion. In revising the text, we have sought to avoid any possibility of readers perceiving disciplinary conflicts, overlooking certain results, or underestimating the contributions of others. We sincerely apologize if the previous version may have given rise to any such misunderstandings.

2. Adaptation: I appreciate the response from the authors regarding their view on adaptation, however I still think that adaptation is defined and used here based on the subjective opinion of the authors and not on an established fact. Perhaps authors are confusing adaptation with memory? I see their argument regarding the use of feedback loop to induce memory-like effects, however this effect remains highly specialized/specific, limited and constrained by various experimental factors, for example, in the gripper demonstrations the operators have to be precisely placed with respect to the laser beams and therefore the introduction of the local stimulus has to be very precise too. These limitations must be clearly discussed along with the generality of these results in the context of adaptation. The authors mention that

reproducibility in such soft matter systems remains limited and that there is a drift in their dynamics over time, given this comment, is the frequency “adaptation” experiment reproducible? If not, is this experiment meaningful?

Our answer: The definition of adaptation in artificial materials remains a topic of ongoing debate. Having worked in the field of life-like materials for many years, our intention in this manuscript was not to provide a comprehensive definition or to engage deeply in this debate. In our view, adaptation is inherently linked to memory, and one effective mechanism to generate memory – illustrated in this study – is the use of a positive feedback loop. We have revised the text to more clearly introduce our working definition of adaptive materials.

Here, we echo the expression by Fratzl, et al, who described adaptive materials as representing a higher level of responsiveness, “the stimulus-induced change in the material encounters a competing reaction and the output results from balancing the two via mutual feedback”.

We are grateful for the Reviewer’s suggestion. In the revised manuscript, we have added a discussion of the limitations of these proof-of-concept demonstrations of adaptation. The revised text now reads:

The gripping and cascading devices illustrated in Fig. 4 serve to demonstrate adaptation at the proof-of-concept level. Their operation requires precise optical beam alignment and accurate positioning of the heat stimulus source within a specific range to overcome the energy barrier governed by the positive feedback loop. For broader applicability, system-level integration between the actuator and the optics will be necessary.

Regarding the issue of “reproducibility in such soft matter systems”, we have thoroughly clarified why soft actuators cannot be treated in the same way as metallic cantilevers (System limitations Section). We further emphasize that our experimental goal is not to achieve precise control or fine-tune oscillatory properties across different samples. In the frequency adaptation experiment, we deliberately designed the system to exhibit two distinct oscillation frequencies – one high and one low. The transition between these frequencies is fully reproducible, as demonstrated in the switching experiments shown in Supplementary Figure 21. We would like to notice that a material state can be characterized not only by its spatial configuration (the shape morphing) but also by its oscillation frequency in the time domain. The purpose of the experiment is to demonstrate that a single oscillator can actually exhibit two distinct frequencies driven by separate feedback loops – not to precisely replicate oscillation behavior across samples, nor to stabilize drift during a single oscillation cycle.

3. Robustness to external perturbations: Perhaps the authors misunderstood my previous comment on robustness of oscillators dynamics. I can see from the figures that the oscillators return (almost always) to the original state after the external perturbation is removed. However, my question was related to the robustness in terms of amount of such experiments that can be performed before the oscillator dynamics is significantly changed and the coupling is lost (if at all). The reported experiments contain only one perturbation cycle and are run for a maximum of 5-6 mins at best. Do the oscillator dynamics and most importantly the light-based interactions persist after long time and after repeated perturbations?

Our answer: We apologize for any misunderstanding. In our design, this phenomenon is obvious: the light beam interaction must persist in order to close the feedback loop. In other

words, as long as the system (two operators) is self-oscillating, the light beam interaction is necessarily present.

To further illustrate this, we present data from an experiment in which two coupled operators (as shown in Fig. 3a–f) were intentionally disturbed by mechanically stopping one of the operators, while monitoring the displacement of the other. The experiment was conducted over the course of one hour, during which the operator underwent three manual cessations followed by self-recovery in each minute – amounting to approximately 180 perturbations in total. Although some drift is observed in the oscillation data, the system consistently remained out of equilibrium, and the optical feedback loop was maintained throughout.

4. Generality and scalability: *The demonstration of light-based coupling in other materials is impressive, however the claim that this strategy does not rely on specific material property seems too broad and unfair and should be alleviated because the coupling is shown for a few soft and light-responsive materials only. Regarding the scalability of this approach, Can the light-based coupling be extended to many operators? Can many operators simultaneously interact with each other (still in pairwise fashion)? Do you need a laser for every operator? If yes, then how does this affect the scalability of this approach to a relatively large network?*

Our answer: Regarding generality, we intentionally categorized the responsive materials encountered in everyday life into two groups: deformable and non-deformable. The aluminum foil–Scotch tape bilayer represents the deformable category, capable of exhibiting both positive and negative feedback, while parafilm represents the non-deformable category, eligible for positive feedback. We employed both materials to realize closed feedback loops for light-interactive operator functions under non-equilibrium conditions. Importantly, the light coupling strategy does not rely on specific material properties, but only requires that the materials exhibit reversible stimulus–response behavior. We therefore believe it is reasonable to state that the demonstrations address the generality of the concept to a considerable extent.

Regarding scalability, our answers are as follows: yes, the concept can be extended to many operators; yes, multiple operators can simultaneously interact in a complex manner; and yes, each operator requires a dedicated light beam. We have added a discussion on system scalability in the revised version, which now reads:

A miniaturized cantilever inherently exhibits higher oscillation frequencies due to fundamental scaling laws. Downscaling self-oscillators enables both higher device density and increased sampling rates. Two-photon polymerization is a well-established fabrication technique for realizing microscopic LCEs and optical waveguides for efficient coupling. We envision scalability arising from the ever-increasing unit density and complexity of the feedback network. Further details see in Supplementary Note 2.5 Scalability – a future plan.

Details of the research plan for systematization can be found in Supplementary Note 2.5.

Finally, we are sincerely grateful to all the Reviewer for the time and valuable feedback, which has significantly helped us improve the quality of our work.